



# Aerosol-stratocumulus interactions: Towards a better process understanding using closures between observations and large eddy simulations

Silvia M. Calderón[1], Juha Tonttila[1], Angela Buchholz[2], Jorma Joutsensaari[2], Mika Komppula[1], Ari Leskinen[1,2], Liqing Hao[2], Dmitri Moisseev[3,4], Iida Pullinen[2], Petri Tiitta[1], Jian Xu[5], Annele Virtanen[2], Harri Kokkola[1], and Sami Romakkaniemi[1]

[1]Atmospheric Research Centre of Eastern Finland, Finnish Meteorological Institute, P.O. Box 1627, 70211 Kuopio, Finland
[2]Department of Applied Physics, University of Eastern Finland
[3]Institute for Atmospheric and Earth System Research/Physics, Faculty of Science, University of Helsinki, Helsinki, Finland
[4]Finnish Meteorological Institute, Helsinki, Finland
[5]Institute of Energy and Climate Research, IEK-8: Troposphere, Forschungzentrum Jülich GmbH, 52425 Jülich, Germany

**Correspondence:** S. Calderón (silvia.calderon@fmi.fi)

**Abstract.** We carried out a closure study of aerosol-cloud interactions during stratocumulus formation using a large eddy simulation model UCLALES-SALSA and observations from the 2020 cloud sampling campaign at the Puijo SMEAR IV station in Kuopio, Finland. The unique observational setup combining in situ and cloud remote sensing measurements allowed a closer look into the aerosol size-composition dependence of droplet activation and droplet growth in turbulent boundary

layer driven by surface forcing and radiative cooling. UCLALES-SALSA uses spectral bin microphysics for aerosols and hydrometeors and incorporates a full description of their interactions into the turbulent-convective radiation-dynamical model of stratocumulus. Based on our results, the model successfully described the probability distribution of updraft velocities and consequently the size dependency of aerosol activation into cloud droplets, and further recreated the size distributions for both interstitial aerosol and cloud droplets. This is the first time such a detailed closure is achieved not only accounting for activation

of cloud droplets in different updrafts, but also accounting for processes evaporating droplets and drizzle production through coagulation-coalescence. We studied two cases of cloud formation, one diurnal (24 September 2020) and one nocturnal (31 October 2020), with high and low aerosol loadings, respectively. Aerosol number concentrations differ more than 1 order of magnitude between cases and therefore, lead to cloud droplet number concentration (CDNC) values which range from less than $100\,\mathrm{cm}^{-3}$ up to $1000\,\mathrm{cm}^{-3}$. Different aerosol loadings affected supersaturation at the cloud base, and thus the size of aerosol

particles activating to cloud droplets. Due to higher CDNC, the mean size of cloud droplets in the diurnal-high aerosol case was lower. Thus, droplet evaporation in downdrafts affected more the observed CDNC at Puijo altitude compared to the low aerosol case. In addition, in the low aerosol case, the presence of large aerosol particles in the accumulation mode played a significant role in the droplet spectrum evolution as it promoted the drizzle formation through collision and coalescence processes. Also, during the event, the formation of ice particles was observed due to subzero temperature at the cloud top. Although the modeled

number concentration of ice hydrometeors was too low to be directly measured, the retrieval of hydrometeor sedimentation velocities with cloud radar allowed us to assess the realism of modeled ice particles. The studied cases are presented in detail





and can be further used by the cloud modellers to test and validate their models in a well-characterized modelling setup. We also provide recommendations on how increasing amount of information on aerosol properties could improve the understanding of processes affecting cloud droplet number and liquid water content in stratiform clouds.

*Copyright statement.* TEXT

# 1 Introduction

Stratocumulus are low-level clouds and therefore respond quickly to changes in boundary layer conditions, especially to perturbations in aerosol properties affecting both, the cloud optical properties and precipitation formation (e.g. Portin et al., 2014; Toll et al., 2019; Eirund et al., 2019; Christensen et al., 2020). From the practical perspective, they provide an excellent way

to study aerosol-cloud interactions as they can be continuously monitored in measurement stations where in-cloud conditions occur frequently. In such clouds, droplets are formed at the cloud base in updrafts, where the updraft strength together with the condensation sink on particles, define the maximum supersaturation that can be reached inside a rising parcel of air, and with that, the fraction of aerosol particles that can activate as cloud droplets (Pruppacher and Klett, 2010). The relative importance of aerosol concentration and updraft strength on droplet number concentration varies and depends on the local conditions,

droplet formation can be characterized to be aerosol or updraft limited in extreme cases, whereas typically both factors contribute (Reutter et al., 2009; Chen et al., 2016, 2018a). From the meteorological point of view, the diurnal variability in the updraft strength is characteristic of stratocumulus and constitutes the dominant variable of cloud dynamics. At the top of the stratocumulus, radiative cooling produces negatively buoyant plumes, downdrafts, that are balanced by updrafts or positively buoyant fluxes of energy and moisture from the surface. The strength of these turbulent circulations is further enhanced by

the gas-liquid energy exchange during condensation processes in updrafts and evaporation and cooling in downdrafts (Wood, 2012). As both radiative cooling strength and surface heat fluxes depend on the amount of solar radiation, this turbulent circulation mixing shows diurnal variability. Between daytime and nighttime, the standard deviation of the vertical wind distribution ($\sigma_w$) can vary from $1\,\mathrm{m\,s^{-1}}$ to $0.3\,\mathrm{m\,s^{-1}}$ (Bougiatioti et al., 2020) with modal values of $\pm\,1\,\mathrm{m\,s^{-1}}$ for updrafts or downdrafts, $w$ (Ghate et al., 2010). Previously $\sigma_w$ has been identified as a key driver of droplet formation and temporal variability of cloud

droplet and ice number concentrations (Sullivan et al., 2016; Bougiatioti et al., 2020). Although in polluted conditions with high aerosol loading, the droplet number concentrations can be even more sensitive to $w$ than to the aerosol composition or even the aerosol number concentration (Donner et al., 2016; Bougiatioti et al., 2020; Kacarab et al., 2020).

The effects of updraft variability on cloud droplet number concentration (CDNC) and shape of cloud droplet size distributions are not only constrained to the droplet activation process at the cloud base. Boundary layer dynamics affect the droplet

spectrum in the cloud domain. In downdrafts, supersaturation in air parcels decreases leading to a reduction in the mean droplet size or even to a complete evaporation of the smallest cloud droplets. The same can also happen at the cloud edges, where entrainment mixing decreases the liquid water content (e.g. Moeng, 2000; Stevens, 2002). Within a cloud, ascending and





descending air particles are mixed with each other making the resulting droplet size distribution broader than the original ones (Hsieh et al., 2009). Beyond, small scale turbulent fluctuations strengthen the size dependency of processes such as evapora-

tion/condensation through the so-called enhanced Ostwald ripening effect (Hagen, 1979) with significant effects on the shape of droplet distributions and thus on hydrometeor growth. For example, it can affect the first steps of precipitation formation through coagulation-coalescence which is highly dependent on the droplet mean size and width of the droplet size distribution (Çelik and Marwitz, 1999; Wood et al., 2002; Romakkaniemi et al., 2009; Yang et al., 2018).

Even with a very good understanding at the process level, the role that turbulent mixing plays in stratocumulus cloud

dynamics is difficult to assess. During the convective overturning, cloud microphysical properties change over time through the cloud domain, thus in situ and remote sensing observations can only provide long-term-single altitude or time-limited-variable altitude data sets. Despite some successful attempts to reconcile observed and predicted droplet number concentration based on cloud condensation nuclei (CCN) concentrations from aerosol activation parameterizations or adiabatic air parcel models (Conant et al., 2004; Meskhidze et al., 2005; Fountoukis et al., 2007), other closure studies have reported an almost

50% overestimation in CDNC in the case of stratocumulus clouds (Snider et al., 2003; Romakkaniemi et al., 2009). The agreement is found to improve after accounting for the entrainment (Morales et al., 2011) or in-cloud evaporation of cloud droplets (Romakkaniemi et al., 2009). The majority of these closure studies have been focused on the aerosol-droplet transition based exclusively on the predominant role of aerosol number concentrations. Closure studies that scrutinize the relationship between simulated in-cloud vertical velocity distributions to observations of droplet size and number concentrations are scarce

(Sullivan et al., 2016; Donner et al., 2016; Bougiatioti et al., 2020; Zhu et al., 2021; Georgakaki et al., 2021). Likewise, large-eddy-simulations oversimplify the aerosol chemical effects during aerosol-cloud-interactions to keep the model complexity in a manageable level. Closure studies based on the more commonly used bulk microphysical models, simulate the cloud droplet spectrum variability but only as deviations from a predetermined droplet size distribution that may be representative of a certain cloud type and atmospheric background conditions, but it is totally or partially disconnected to those aerosol chemical effects

that control the water balance at the droplet surface (Schemann et al., 2020; Stevens et al., 2020).

Besides the effect on the aerosol-CCN-droplet transition, it is necessary to explore how in-cloud turbulent convection modulates droplet size and number concentrations through changes in other microphysical processes such as droplet depletion by collision-coalescence during drizzle and precipitation formation, as well as by evaporation during mixing with cloud-free air after lateral and vertical entrainment. Since these processes affect the relationship between droplet properties at the cloud base

and the cloud top, they have been pointed out as key issues to improve the retrieval of CCN and CDNC properties using ground-base and satellite remote sensing data (Quaas et al., 2020). Here, we have addressed some of these issues by performing a study on aerosol-cloud interactions in stratocumulus clouds involving detailed modelling of aerosol size and composition effects on cloud microphysical processes with a large-eddy-simulation model UCLALES-SALSA model (University of California Los Angeles Large Eddy Simulation model-Sectional Aerosol module for Large Applications) (Tonttila et al., 2017).

Modelling results are compared to a unique observational setup comprising time series of altitude-dependent distributions of the vertical wind velocity, activation efficiency curves, aerosol and droplet size and number concentrations, and radar velocity



distributions. Observations were carried out during the 2020 sampling campaign at the Puijo SMEAR IV station in Kuopio, Finland as part of the measurement campaigns within the FORCeS Project.

We studied two cases of stratocumulus cloud formation: one diurnal case on 24 September 2020 and one nocturnal case on
31 October 2020 with high and low aerosol loadings, respectively. Aerosol number concentrations differ more than an order of magnitude between cases and therefore, lead to droplet number concentrations of less than $100\,\mathrm{cm}^{-3}$ up to $1000\,\mathrm{cm}^{-3}$. This allowed us to gain a deeper understanding of the covariance effect of aerosol loadings and vertical wind variability on droplet number concentrations observed in other studies (e.g. Kacarab et al., 2020; Bougiatioti et al., 2020). We also performed a model sensitivity analysis to explore the significance of aerosol number concentration, mixing state, and ice formation potential on
the cloud droplet microphysics of stratocumulus clouds. These Puijo cloud events can be used by the research community as study cases of stratocumulus formation in boreal environments with anthropogenic influence and additional effects of biomass burning emissions.

## 2  Methods

### 2.1  UCLALES-SALSA modelling framework

UCLALES-SALSA is a large eddy simulation model with explicit calculation of microphysical processes of aerosol particles and hydrometeors (Tonttila et al., 2017; Ahola et al., 2020; Tonttila et al., 2021). Dynamics of the atmospheric boundary layer are represented with UCLALES, University of California Los Angeles Large Eddy Simulation model (Stevens et al., 2005) while the dynamics of aerosol and hydrometeor populations are represented with SALSA, Sectional Aerosol module for Large Applications (Kokkola et al., 2008; Tonttila et al., 2021). In this way, UCLALES-SALSA is a versatile modelling
framework that allows for studying irradiance changes caused by aerosol-radiation and aerosol-cloud interactions with small-scale meteorology. Previous applications of the model include studies on the aerosol-radiation feedback in cloud-free boundary layers (Slater et al., 2020), the cloud-radiation feedback in marine stratocumulus-capped boundary layers (Tonttila et al., 2017), Artic ice and mixed-phase clouds (Ahola et al., 2020), and fog events (Boutle et al., 2018, 2022), and cloud seeding mechanisms for the artificial enhancement of precipitation (Tonttila et al., 2021). As shown in these studies, with this modelling
framework, we can perform a full closure study of aerosol-cloud interactions studying in detail how the updraft velocity distribution modulates the droplet activation process through the interplay between aerosol size and number concentrations and supersaturation values. Also, how the strength of convective circulation affects the shape of the cloud droplet size distribution through changes in evaporation-condensation and collision-coalescence rates.

UCLALES (Stevens et al., 2005) resolves time series of the wind vector field and scalar fields of potential temperature
and total water mixing ratio in a tridimensional model domain where sub-grid scale turbulent fluxes are modeled with the Smagorinsky-Lilly parameterization (Smagorinsky, 1963). Radiative fluxes are modeled with the $\delta$-four stream radiative transfer code of Fu and Liou (1993) as modified by Stevens et al. (2005). Horizontal boundary conditions are doubly periodical and fixed in the vertical direction. Advection of momentum variables is represented by a fourth-order difference equation with time stepping and numerically solved by leapfrog integration. The model uses a damping layer at the top of the domain to control





unwanted gravity waves (Stevens et al., 2005; Tonttila et al., 2017, 2021). The large-scale subsidence is calculated assuming
uniform divergence to assure balance between subsidence warming and radiative cooling above the inversion (Stevens et al.,
2005; Ackerman et al., 2009). Surface topography is not directly taken into account, instead of surface sensible and latent heat
fluxes are given as an input or calculated using the coupled soil moisture and surface temperature scheme by Ács et al. (1991).

SALSA (Kokkola et al., 2008, 2018) uses spectral bin microphysics to represent the properties of aerosol particles and
cloud hydrometeor in the atmosphere including processes for aerosol particle and hydrometeor growth or shrinkage by water
condensation or evaporation-sublimation, hydrometeor growth via collision-coalescence (i.e. accretion), droplet activation via
cloud condensation nuclei or ice nuclei, aerosol formation via gas to particle conversion, and aerosol scavenging via collision-
coalescence. The model can simulate ice formation via homogeneous freezing at temperatures below -30 °C or via heteroge-
neous freezing at higher temperatures through immersion and deposition mechanisms. Riming and ice aggregation are also
considered (Ahola et al., 2020; Tonttila et al., 2022). During all these processes the mass/number size distributions of aerosol
particles are tracked as presented in Tonttila et al. (2017, 2021). Aerosol particles can be represented either as externally mixed
or internally mixed populations. Chemical composition effects are accounted for during cloud droplet activation in solving con-
densation of water to aerosol and cloud hydrometeors and during ice nuclei formation using water activity and contact angle
distribution to describe heterogeneous ice nucleation efficiency (Khvorostyanov and Curry, 2000; Ahola et al., 2020; Tonttila
et al., 2022). Aerosol particles are separated into non-activated and activated particles depending on water supersaturation and
size of particles, and then redistributed among size bins between interstitial aerosol and cloud droplets. More information about
aerosol size and composition and bin schemes can be found in the original SALSA description by Kokkola et al. (2008, 2018);
Tonttila et al. (2017); Ahola et al. (2020); Tonttila et al. (2021). Microphysics of liquid droplets was explained by Tonttila et al.
(2017, 2021) while ice microphysics was described by Ahola et al. (2020); Tonttila et al. (2022). Section 1 of the supporting
information includes details of modelling frameworks used for each one of the microphysical processes.

## 2.2    in situ measurements during Puijo 2020 campaign

The Puijo 2020 campaign was carried out at the Puijo SMEAR IV station in Kuopio, Finland (62.9092°, 27.6556°, 306 m
above mean sea level, 225 m above local lake level) between September 15th-November 30th 2020. It is one of the measure-
ment campaigns within the FORCeS Project (European Union's Horizon 2020 research and innovation programme under grant
agreement No 821205, 2019). The Puijo station has been active since 2006 providing continuous observations on meteorolog-
ical parameters, aerosol size distributions and optical properties, cloud droplet size distributions, and concentrations of trace
gases (Portin et al., 2009). Although the station is at an elevated location at the top of Puijo hill covered by boreal forests 75 m
above ground and approximately 225 m above the surrounding lake level, the effect of local topography on observed cloud
properties is limited to certain high wind conditions (Romakkaniemi et al., 2017). The location is also particularly adequate to
perform long-term continuous measurements of aerosol-cloud interactions since cloudy conditions are observed at the station
approximately 8% of the time (Ruuskanen et al., 2021). More information about the Puijo station can be found in the literature
(Leskinen et al., 2009, 2012; Portin et al., 2014).





Aerosol number concentrations and size distributions were measured using the Twin-inlet system composed of two differential mobility particle sizer instruments (DMPS) connected in parallel to two separate inlets, from now on labeled as total and interstitial. The heated total inlet measures activated and non-activated particles with a diameter below $40\,\mu m$ (DMPS-total).
The interstitial inlet measures concentrations of particles with diameter equal to or lower than $1\,\mu m$ considered as non-activated or interstitial aerosol (Conant et al., 2004), that have been previously separated with a $PM_1$-impactor (DMPS-interstitial). The number concentration of activated droplets is calculated as the difference between the number concentrations of the total and interstitial lines in the size range from $28\,nm$ to $800\,nm$ and from $28\,nm$ to $560\,nm$, respectively. Activation efficiency curves were retrieved from these observations using the activated fraction as a function of dry particle size calculated as the ratio between activated particles and total particles (activated + non-activated) in a size bin. More details about the Twin-inlet DMPS system can be found in literature (Portin et al., 2009, 2014; Ruuskanen et al., 2021). At Puijo, the Twin-inlet DMPS system has been successfully employed in studies related to size-dependent activation of aerosol particles and partitioning of different chemical components between the interstitial aerosol particles and cloud droplets (Hao et al., 2013; Portin et al., 2014; Väisänen et al., 2016; Ruuskanen et al., 2021).

The bulk chemical composition of non-refractory $PM_1$ aerosol particles was measured with an Aerosol Chemical Speciation Monitor (ACSM) (Ng. et al., 2011) to yield the contribution of sulfate, nitrate, ammonium, and organic species. The mass size distribution of these species was measured with a High-Resolution Time-of-Flight Aerosol Mass Spectrometer (HR-ToF-AMS, Aerodyne Research Inc.) (DeCarlo et al., 2006) located at a nearby station at the foot of Puijo hill.

Droplet number concentrations and size distributions were measured using the forward-scattering optical spectrometer (Fog Monitor) described by Spiegel et al. (2012) (FM-120, Droplet Measurement Technologies Inc., USA) with an observation range of 30 bins from $2\,\mu m$ to $50\,\mu m$. Additionally, the number concentration and size distributions of large droplets, and ice particles were measured with the holographic imaging system (Icing Condition Evaluation Method, ICEMET) described by Kaikkonen et al. (2020) with an observational range from $5\,\mu m$ to $200\,\mu m$ (Tiitta et al., 2022).

All instruments, except the AMS, were located in the Puijo station at the top of the tower. The AMS instrument was located at ground level approximately 200 m below tower altitude. The small difference in altitude leads us to assume that measurements from all instruments correspond to the same air parcel, and therefore, are representative of atmospheric conditions.

To complement our observational data set, we used information available for two measurement sites nearby, the Savilahti and Vehmasmäki stations. The Savilahti station is located in a semi-urban environment, ca. $2\,km$ southwest of the Puijo SMEAR IV station (5 m above the surrounding lake level). It has an automatic weather station that operates regularly to provide 1 min resolution data of air and ground temperature, relative humidity, wind speed, and wind direction and pressure as well as cloud base height using a ceilometer (Vaisala CT25K). Meteorological data from the Savilahti station are representative of Puijo conditions due to the proximity between stations. During the campaign, Savilahti station also provided observations for wind profiling that was useful to assess the ability of the model to describe the vertical wind distribution. Vertical profiles of the vertical wind velocity at altitudes up to $11\,km$ were retrieved from observations taken by a Doppler radar–radiometer system (94-GHz dual-polarization frequency-modulated continuous-wave Doppler cloud radar HYDRA-W) described by Küchler et al. (2017). In addition, vertical wind velocity at the cloud base was retrieved from observations of a Doppler lidar (Light





Detection And Ranging, Halo Photonics) described by Tucker et al. (2009). The operational scanning strategy and calculation methods used to detect cloud conditions from Doppler lidar measurements are explained by Hirsikko et al. (2014) and Manninen et al. (2018). Doppler lidar wind velocities were used to study cloud base conditions when the lowest retrieved height with observable cloud-driven turbulence was above the lowest observable Doppler lidar range gate of 105 m (Manninen et al., 2018) and also equal to or higher than the cloud base height detected with the ceilometer. The lowest observed altitude of 105 m was also used in the analysis of cloud base updraft velocity if the cloud base was below this limit. Data sets from these instruments are available from the Aerosol, Clouds and Trace Gases-ACTRIS data centre (CLU, 2022).

Vertical profiles of temperature, wind speed and wind direction as well as specific humidity and pressure were obtained from the tall mast at the Vehmasmäki station. This station is located in a forested rural area, 13 km southwest to the Puijo station. This station operates regularly and provides time series with 1 min resolution of the vertical profiles of meteorological variables, temperature and relative humidity up to 300 m above ground , wind velocity, and direction up to 272 m above ground.

Section 2 of the supporting information provides data relevant to the instrumentation used in this closure study.

## 2.3 Cloud events during the Puijo 2020 campaign

A cloud event was defined as a continuous time period, longer than 1 hour (Väisänen et al., 2016) during which observations at the Puijo top station met the criteria of cloudy conditions established as liquid water content above $0.01\,\mathrm{g\,m^{-3}}$, cloud droplet number concentration higher than $50\,\mathrm{cm^{-3}}$ and visibility values below 200 m on average. During the Puijo 2020 campaign, there were 49 cloud events, 20 of them during day time. We selected two cloud events where cloud boundaries were well defined by radar and lidar observations to study aerosol-cloud interactions in detail by combining observational data and LES modeling. Selected events reflect contrasting scenarios of cloud formation in terms of the aerosol loading and turbulence driving mechanism. Cloud properties and other relevant data about the aerosol number and mass concentration and aerosol chemical composition are summarized in Table 1. More details are included in Section 3 of the supporting information.

## 2.4 Model setup

The model domain comprised a horizontal grid of 64 by 64 equidistant points with a vertical grid extended up to an altitude equivalent to three times the cloud top height retrieved from radar profiles. This assures that the model domain has enough space above cloud layer to capture the dynamics of large-scale processes associated to instability at the entrainment zone in the cloud top (Mellado, 2017). Vertical grid spacing was set at 10 m as no significant changes in model outputs were observed when finer resolution was employed. Differential equations were resolved using an Eulerian-Lagrangian time-stepping method with a maximum time step of 0.5 s (Case 1) or 1 s (Case 2). A shorter time step was used for case 1 to minimize the appearance of spurious supersaturation values at the cloud top that are commonly observed in large-eddy simulations (Stevens et al., 1996; Grabowski and Morrison, 2008; Hoffmann, 2016). Since the model can describe the influence of the diurnal cycle of solar insolation via solar zenith angle, the latitude as well as the time were carefully defined to match conditions at the station. Latitude at the Puijo station was set to be 62.53 degrees. Simulations were started two hours before the beginning of the period of interest, the first hour was set as a spin-up period to allow the turbulence to develop in the absence of collision



**Table 1.** Cloud and aerosol properties during selected cloud events that were measured at the Puijo top monitoring site. Values are reported as an arithmetic mean $\pm$ standard deviation (number of observations). $N_{tot}$ and $N_{acc}$ are aerosol number concentrations in the total size range from 27 nm to 1000 nm and in the accumulation mode from 100 nm to 1000 nm, respectively. CDNC represents droplet number concentration retrieved from Twin-inlet DMPS measurements

| Cloud event | 24 September 2020 | 31 October 2020 |
|---|---|---|
| Time, UTC+02:00 | 07:54 - 12:49 | 00:35 - 06:35 |
| [a]Retrieved cloud base height [m] | $63 \pm 39$ (296) | $125 \pm 42$ (360) |
| [b]Retrieved cloud top height [m] | $357 \pm 56$ (6436) | $457 \pm 23$ (5588) |
| [c] $N_{tot}$ [ cm$^{-3}$] | $2042 \pm 110$ (5) | $164 \pm 102$ (6) |
| [c] $N_{acc}$ [ cm$^{-3}$] | $1347 \pm 46$ (5) | $80 \pm 43$ (6) |
| [d] CDNC [ cm$^{-3}$] | $417 \pm 211$ (3486) | $86 \pm 23$ (3394) |
| [e] CDNC [ cm$^{-3}$] | $687 \pm 164$ (5) | $87 \pm 50$ (6) |
| Model parameters related to SALSA: aerosol size distribution used in base simulation | | |
| Mode aerosol number concentration[f] [mg$^{-1}$] | [879,1325] | [456, 155, 25] |
| Mode geometric mean diameter [μm] | [0.076, 0.156] | [0.039, 0.215, 0.735] |
| Mode standard deviation | [1.8205, 1.464] | [1.5249, 1.5826, 1.1811] |
| Dry particle composition in volume fraction | [0.255 SO$_4$, 0.745 OC] | [0.12 SO$_4$, 0.88 OC] |

[a] Ceilometer, [b] Cloud radar [c] Twin-inlet differential mobility particle sizer (Twin-inlet DMPS), total inlet. [d] Fog Monitor FM-120

[e] Retrieved from Twin-inlet DMPS system as the concentration difference between the total and interstitial lines

[f] Expressed per mass unit of moist air as required by UCLALES-SALSA

processes and drizzle formation, which were allowed for the second hour before the actual analysis started (Tonttila et al., 2017). Time-series of surface temperature measured at the Savilahti station were fitted into a time-dependent function. This equation was introduced into the UCLALES-SALSA model to calculate the corresponding changes in the surface fluxes of latent and sensible heat in the simulation of Case 1.

Initial conditions for UCLALES-SALSA simulations were set by using vertical profiles of potential temperature, specific humidity and horizontal wind components taken from reanalyzed data from ECMWF-ERA5 (Hersbach et al., 2020) and meteorological data from stations in the proximity of Puijo tower. Data from the Savilahti station, the closest to Puijo, were used for surface conditions. Being apart from the Puijo station, data from the Vehmasmäki mast were considered to represent atmospheric background conditions during cloud events. The location and strength of the inversion layer were found by comparison
of temperature mast observations, cloud radar information on cloud top altitude, and reanalyzed vertical temperature profiles from ECMWF-ERA5 data. The reanalyzed data were used to augment profile data at higher altitudes where observations were not available (Hersbach et al., 2020).

    To calculate atmospheric radiative transfer, the simulations also require background profiles including temperature, specific humidity, and ozone concentrations at pre-defined pressure levels going from 1000 Pa to 1 Pa. These data were retrieved from





the ECMWF-ERA5 data set " hourly data on pressure levels from 1979 to present" for the time corresponding to the beginning of the cloud event using 27.61 degrees and 62.90 degrees as longitude and latitude, respectively.

Initial conditions for size-segregated aerosol number concentrations were fed into the model as multimodal lognormal functions $n_N(D_p)$ with parameters fitted to measurements taken with the Twin-inlet DMPS system from the total inlet at the beginning of each cloud event, 24 September 2020 07:54 (UTC+2) and 31 October 2020 00:35 (UTC+2). Parameters for size

distributions are reported in Table 1. Aerosol particles were assumed to be internally mixed. Aerosol main constituents were sulfate ($SO_4$) and organic carbon (OC) species. We used the term organic carbon species as a simplification of the denomination of "organic aerosol". Aerosol particles were assumed to have a density equivalent to the material density or molar fraction weighted average of individual densities as pure solid (DeCarlo et al., 2004). Density values used for calculations and additional details about the aerosol composition are included in Section 4 of the supporting information.

In the base scenario of aerosol composition, identified here as internally mixed aerosol, all particles have the same composition. The particle composition in volume fraction was retrieved from the event-average mass size distributions measured by the AMS. Calculations involved are included in Section 4 of the supporting information. For the simulation of the mixed-phase cloud case, we changed the representation of aerosol composition to an externally mixed population composed of two regimes, A and B, both with the same aerosol size distribution shape. While regime A was composed of $SO_4$ and OC, dust was incor-

porated as an aerosol constituent of particles in regime B to provide ice nucleating particles. Number concentrations and exact composition are reported later in the analysis of the cloud case.

Reported values of mean contact angle for natural dust vary widely (e.g. Chen et al., 2008; Hoose et al., 2010; Kulkarni and Dobbie, 2010; Wang et al., 2014; Savre and Ekman, 2015) and there is no consensus on how to parameterize its ice nucleation ability. In the lack of experimental information about the ice nucleation ability of our aerosols, we assumed a contact angle of

$79\,^\circ \pm 12\,^\circ$ inside the range of variation observed for proxies of atmospheric mineral dust such as kaolinite, illite and quartz coated with sulfuric acid (Knopf and Koop, 2006; Chernoff and Bertram, 2010; Murray et al., 2012).

Closure studies of cloud properties are particularly challenging due to the spatial variability of cloud dynamics since averaging operations across the model domain can mask important correlations between cloud properties on the micro and macro scales. Although observations are subject to the same variability, any conclusion derived from the degree of agreement between

model results and observations must be evaluated carefully. Detailed explanations about the treatment of model outputs (e.g. averaging operations across model domain) and observations are included in Section 5 of the supporting information.

## 3 Results

During the second sampling week of the Puijo campaign, between 24 September 2020 and 10 October 2020, observations showed aerosol mass concentrations and aerosol contents of organic and black carbon that were higher than long-term average

values. Back trajectory analysis in combination with information from the European Forest Fire Information System (EFFIS) (San-Miguel-Ayanz et al., 2012) confirmed that air mass origins were located in areas of central/eastern Europe affected by wildfires (Buchholz et al., 2022). Aerosol mass concentration decreased to long-term average values of clean atmospheric





conditions after 11 October 2020. For the analysis, we selected two well-characterized cloud cases; Case 1 occurring during and Case 2 after this forest fire plume period. This allowed us to investigate the sensitivity of the stratocumulus formation to

aerosol number concentrations. Case 1 corresponds to a cloud event occurring with constant high aerosol loadings from the early morning to noon on 24 September 2020. In contrast, Case 2 is a cloud event that occurred from midnight until early morning on 31 October 2020 with low aerosol loadings that decreased rapidly through the particle size range with time during the event. Cloud radar profiles showed clear sky conditions above cloud top for both cases, which favored studying aerosol-cloud-radiation interactions without interference from higher-level ice clouds which could have affected radiative cooling at

the cloud top (Wood, 2012).

Figure 1 and Figure 2 the atmospheric boundary layer properties during both cloud events, together with the vertical profiles used to initialize our simulations. We used a color scale to link time with the variation of each property. We monitored this variability before and during the cloud event to identify the transition from cloud-free to cloudy conditions.

For the diurnal cloud case or Case 1, Fig. 1 indicates the existence of a $170\,\text{m}$ deep well-mixed boundary layer capped

by an inversion layer $180\,\text{m}$ deep followed by neutral stability conditions at higher altitudes. During the cloud event, the boundary layer showed high moisture contents with relative humidity ranging from 99% to 90% at the surface. Observed profiles indicated that Case 1 started as a fog episode growing in height and transforming into a stratocumulus cloud before complete dissipation as suggested by Radar profiles included in section 5 of the supporting information. We represent a quasi-ideal well mixed boundary layer with a constant total moisture mixture ratio of $7.95\,\text{g}\,\text{kg}^{-1}$ and a potential temperature of

$283\,\text{K}$ in the mixed layer. To capture the observed variability, we applied a moderate temperature increment of $0.2\,\text{K}$ at the inversion base $170\,\text{m}$ with a reduction of the total moisture content to $7.57\,\text{g}\,\text{kg}^{-1}$. Instead of having a sharp jump in the vertical variation of atmospheric properties, we assumed that temperature and moisture vary with constant gradients of $2.3\,\text{K}(10\,\text{m})^{-1}$ and $-0.028\,\text{g}\,\text{kg}^{-1}(10\,\text{m})^{-1}$ from the inversion base up to the inversion top located at $350\,\text{m}$. At higher altitudes, our vertical profiles move towards ERA-5 data since observations were not available. To simulate the horizontal components of the wind

velocity, we interpolated observed vertical profiles from the Vehmasmäki station using data before and during the cloud event. The resulting initial profiles showed constant values for the horizontal components of the wind velocity, u and v respectively, with increasing altitudes up to the inversion base. In terms of aerosol properties, Case 1, started during the smoke plume period and evolved with sustained high aerosol loadings of ca. $2000\,\text{cm}^{-3}$ and dry particle mode diameters of $0.076\,\mu\text{m}$ and $0.156\,\mu\text{m}$ calculated by fitting of DMPS observations to lognormal size distributions. Long-range transport of air masses containing

biomass burning emissions kept high aerosol mass concentrations that did not significantly change during the cloud event as it was reported in Table 1. The aerosol composition was dominated by organic carbon (66 ± 4 % w/w) and sulfate species (34 ± 4 % w/w). The wind direction at the monitoring site does not change significantly during the cloud event. Since this cloud event evolves from early morning until noon, we were able to follow diurnal cloud dynamics induced by solar insolation, i.e. direct response to changes in radiative cooling at the cloud upper region, as well as changes in cloud droplet activation induced

by changes in the turbulence structure caused by increasing surface fluxes of moisture and heat.

Case 2 was nocturnal and lasted for six hours with a stable cloud base and top altitudes at approximately $105\,\text{m}$ and $420\,\text{m}$, respectively. Observations indicated drizzle formation and development of very light snowfall due to subzero temperatures in





the cloud upper section. Values of aerosol mass concentration, almost one-tenth of those observed in the diurnal case, were rapidly and monotonically decreasing with time. Aerosol composition varied more than that in Case 1 with average mass

fraction values of 46 ± 34 % w/w and 55 ± 34 % w/w for organic carbon and sulfate species, respectively. Average mass concentrations of aerosol chemical constituents were in the same order of magnitude as those measured during the Puijo campaigns of 2010 and 2011 for clean atmospheric background in both, clear sky and in-cloud conditions (Portin et al., 2014). This cloud event, therefore, helps to understand the processes in stratocumulus under low aerosol loadings. Unlike the diurnal cloud case, the cloud top rise was limited by a stronger and deeper inversion layer, and the temperature and total moisture

content of the boundary layer were reduced with time and they were lower than those of Case 1. In addition, there was a prominent mode of aerosol particles with mobility diameter above $0.5\,\mu m$ that was not observed in aerosol size distributions during Case 1. Large particles in the sub-micron range promotes drizzle formation (Tonttila et al., 2021). The initial profiles of atmospheric properties used for simulation of Case 2 are shown in Fig. 2. The inversion layer started at $350\,m$ with a temperature jump of $1.3\,K$ from $269\,K$, after which the temperature increased by $2.15\,K(10\,m)^{-1}$ up to $650\,m$, approximately.

Atmospheric stability was assumed at higher altitudes. For the wind profiles, the model was initialized with observed values.

Cloud cases are now discussed separately as each one of them reflects different aerosol-induced effects on cloud microphysical processes. Each case is analyzed in a similar way moving from the macroscopic point of view (i.e. liquid water content, in-cloud vertical wind distribution) to cloud microphysical properties and processes (i.e. aerosol and droplet size distributions, droplet activation efficiency). For both cloud cases there is also a model sensitivity analysis to evaluate changes in cloud

dynamics induced by perturbations of aerosol properties (i.e. mixing state, number, and size distributions).

### 3.1 Case 1: diurnal cloud event with high aerosol loading

### 3.1.1 Cloud boundaries

The comparison of modeled cloud properties to observations starts with macroscopical properties related to cloud base and cloud top boundaries. Figure 3 shows average vertical profiles of cloud liquid water content and cloud droplet number con-

centrations simulated with UCLALES-SALSA for Case 1. Model outputs are presented as horizontal average values in a color scale whose lower limit corresponds to $0.01\,g\,m^{-3}$. Figure 3 also includes time series of experimentally observed cloud base and cloud top heights, as well as observed liquid water content or total droplet number concentrations measured at the altitude of the Puijo station. These values are denoted by colored circles in the same color scale defined for model outputs.

Liquid water content (LWC) can be used to define cloud boundaries. From the modelling point of view, we linked cloudy

conditions to grid points of the model domain where LWC was equal to or above $0.01\,g\,m^{-3}$ (Stevens et al., 2005). From the experimental point of view, the cloud base height was retrieved from ceilometer and Doppler lidar observations, while the cloud top height was retrieved from time-dependent vertical profiles of radar reflectivity (dBZ) measured with cloud radar, all the instruments located at the Savilahti station. Radar profiles can be found in Section 5 of the supporting information.

In Fig. 3 we notice that model outputs for both, liquid water mixing ratio and cloud droplet number concentrations varied

accordingly to observations between cloud boundaries. Liquid water contents inside the cloud domain increase with height

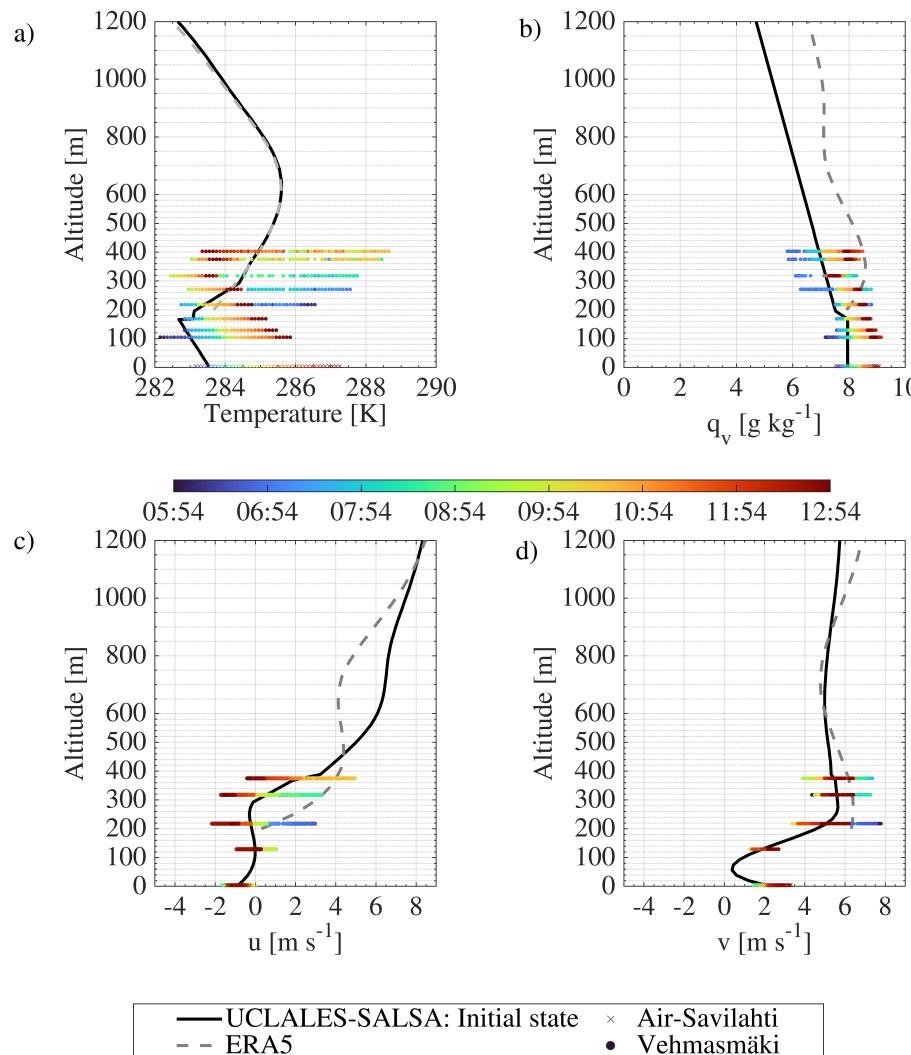

**Figure 1.** Vertical profiles used to initialize the simulation of Case 1, diurnal cloud event of 24 September 2020 starting at 07:54 (UTC+2:00). a) Potential temperature b) Specific humidity c) u-component d) v-component of the horizontal wind velocity. Each panel also shows local surface observational data from the Savilahti station, local vertical profiles observed at the Vehmasmäki station and re-analyzed data from ECMWF-ERA5

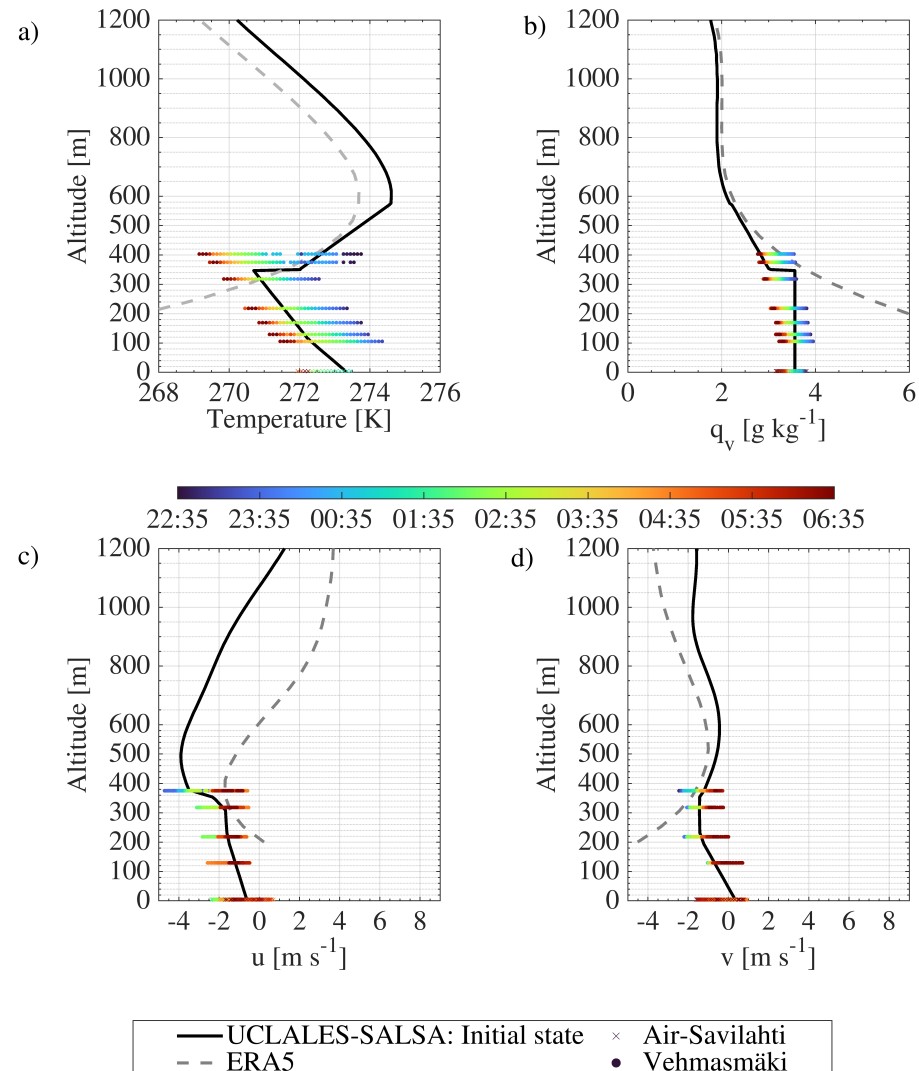

**Figure 2.** Vertical profiles used to initialize the simulation of Case 2, nocturnal cloud event of 31 October 2020 starting at 00:30 (UTC+2:00). a) Potential temperature b) Specific humidity c) u-component d) v-component of the horizontal wind velocity. Each panel also shows local surface observational data from the Savilahti station, local vertical profiles observed at the Vehmasmäki station and re-analyzed data from ECMWF-ERA5





with maximum values at cloud top that are in the order of $0.5\,\mathrm{g\,m^{-3}}$, while cloud droplet number concentrations vary less in the vertical direction and increase with time to up to $1000\,\mathrm{cm^{-3}}$ when calculated in the same observational size range of the fog monitor. Case 1 starts as a fog episode and slowly evolves to a cloud that rises with time in altitude so that the cloud base height rises slowly in the early morning hours and much faster at noon, towards the end of the cloud event. As can be seen, the
observed change in the cloud base height differs quantitatively from the model simulation, and the difference is likely caused by the heterogeneous terrain including nearby lakes that affect both latent and sensible heat fluxes. Due to a lack of information on lake water temperature and small simulation area, we have assumed that it is equal to the land surface temperature for the modeled domain. These factors make a full comparison of model outputs to the full set of observations difficult, as the first two hours might include surface topography effects on cloud dynamics that are not explicitly accounted for by UCLALES-SALSA.
Close to Puijo station, the observed cloud can actually have some characteristics of fog when both the wind speed and cloud base are low. For this reason, the comparison of observations and modelling of Case 1 is focused on the last three hours of the cloud event where there is a significant degree of agreement between observations and model outputs for both, liquid water content and total droplet number concentrations. This time period is marked with gray dotted vertical lines on each panel of Fig. 3.

Stratocumulus capped boundary layers have two distinctive features that correlate to each other, the convective instability driven by cloud-top radiative cooling and the temperature inversion immediately above cloud-top that is maintained by the former (Wood, 2012). In diurnal clouds, this balance is also affected by the incoming solar radiation which warms the surface, causing positive heat flux and that lead to positive buoyancy fluxes. This, in general, tends to increase the turbulence intensity in the whole cloud domain. In our simulation for Case 1, we included a linear increase in the surface temperature equivalent to
one-degree kelvin per hour to simulate the observed surface heating effect caused by solar radiation according to measurements at the Savilahti station.

The average temperature inversion is $7.7\ \mathrm{K\,(100\,m)^{-1}}$ and the cloud-top cooling rate decreases from $68\ \mathrm{W\,m^{-2}}$ to $46$ $\mathrm{W\,m^{-2}}$ at an estimated linear rate of $-2.0\ \mathrm{W\,m^{-2}\,h^{-1}}$ during the cloud event (section 6 of the supporting information). Since aerosol composition and number concentrations do not change significantly during Case 1, the rise in surface-driven convective
mixing produces higher cloud droplet concentrations in the last hours of the event as can be seen in Fig. 3b).

### 3.1.2 In-cloud vertical wind distribution

The increase in turbulent intensity can be followed by comparing the histograms of vertical wind velocities during the cloud event. Figure 4 compares histograms of vertical wind velocities for the third and the fifth hour. Each panel includes model outputs from UCLALES-SALSA and Doppler lidar observations that correspond to the same altitude and time interval. The
altitude at which the wind velocity is retrieved corresponds to the estimated cloud base. Histograms for the remaining hours are included in section 7 of the supporting information.

We used the overlapping index as a statistical measure of agreement between model-based and observation-based distributions of the vertical wind. The overlapping index (OVL) between two different probability distributions that describe the





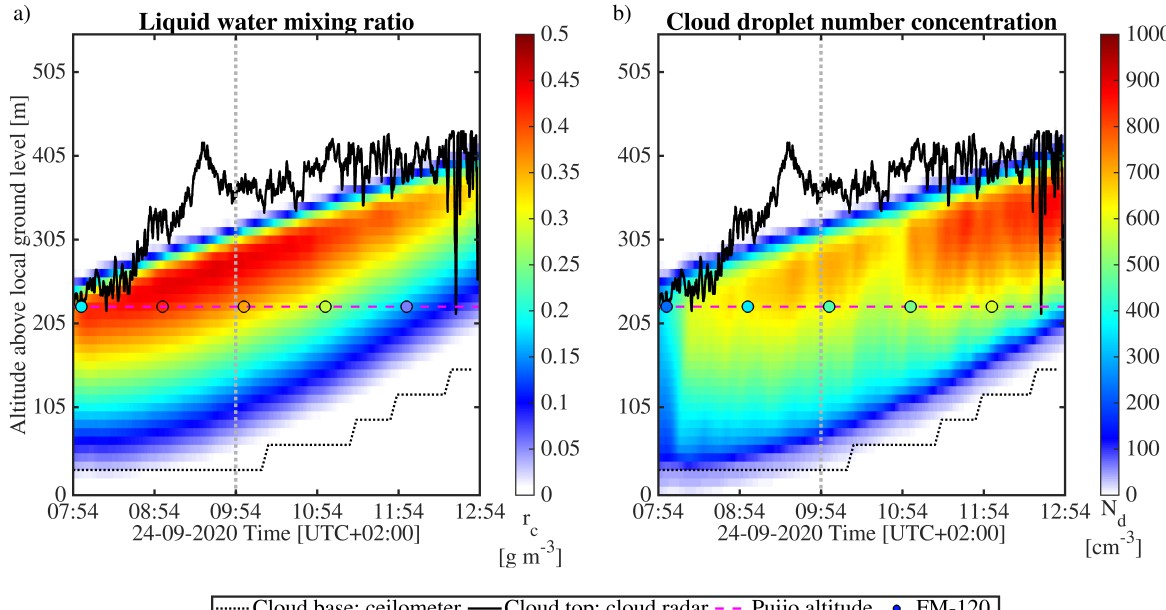

**Figure 3.** Comparison of cloud boundaries for Case 1- 24 September 2020 defined by modeled liquid water content and cloud boundaries retrieved from cloud radar and ceilometer observations. a) modeled vertical profiles of liquid water content and b) model-based cloud droplet number concentrations. Both panels show observations at Puijo altitude from the fog monitor (FM-120). Model-based variables were calculated for the same droplet diameter range of the FM-120. Gray dotted vertical lines mark the third hour of the cloud event in each panel, respectively.

behavior of the same variable x is defined as

$$\mathrm{OVL} = \int min\left[f_1\left(x\right), f_2\left(x\right)\right] dx = \sum min\left[p_1\left(x\right), p_2\left(x\right)\right], \tag{1}$$

where $x$ is the studied variable, in our case, the vertical wind velocity, $f_1\left(x\right)$ and $f_2\left(x\right)$ are the probability density functions (pdf) and $p_1\left(x\right)$ and $p_2\left(x\right)$ are probability distributions of the vertical wind velocity based on observations and modeled by UCLALES-SALSA, respectively (Inman and Bradley Jr., 1989).

In general, the frequency distribution, variance, and skewness of calculated and observed updraft or downdraft winds are in good agreement as reflected by OVL values close to unity. At the third hour the distribution of vertical wind for the cloud base shown in Fig. 4.1 is narrower with the majority of the modeled and observed values between $-0.6\,\mathrm{m\,s^{-1}}$ and $0.6\,\mathrm{m\,s^{-1}}$ with an average standard deviation of $0.4\,\mathrm{m\,s^{-1}}$. When time passes, surface fluxes promote the turbulent mixing increasing the frequency of stronger updrafts/downdrafts. The distribution of the vertical wind broadens out as shown by Fig. 4.2, the model-based hourly average standard deviation increases to $0.5\,\mathrm{m\,s^{-1}}$ at the fifth hour.

While turbulent mixing at the cloud base has a preponderant role in the aerosol-to-droplet transition, it also affects other cloud microphysical processes through changes in the droplet concentration and the shape of droplet size distribution, especially

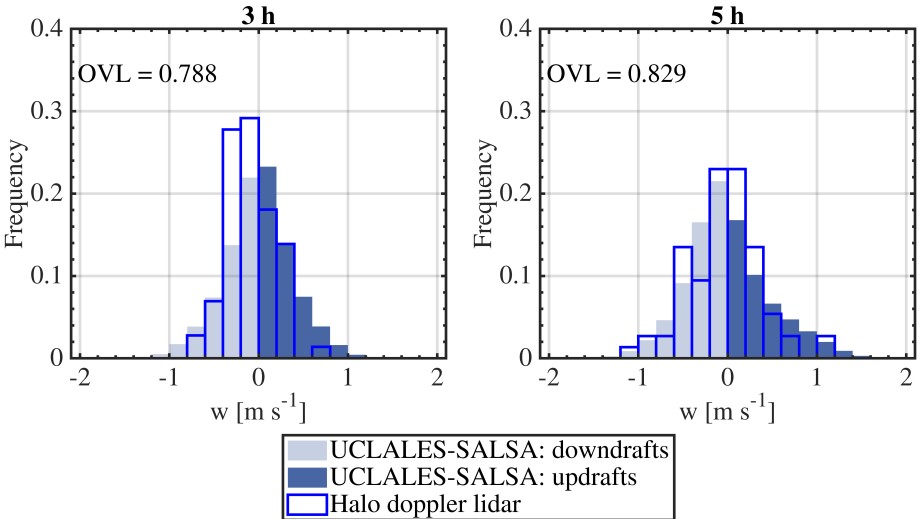

**Figure 4.** Comparison of model-based distributions of vertical wind at cloud base during Case 1 24 September 2020 to those retrieved from Doppler lidar observations. Each panel shows the overlapping index value (OVL) as an indicator of agreement between distributions.

those driven by the collision-coalescence mechanism. To gain insights on turbulent-induced effects inside the cloud domain, we compared the vertical wind distribution using model outputs and observations from the cloud radar.

For the sake of brevity, we did not include here the distribution of the vertical wind for each one of the specific altitudes
inside the cloud domain at which there are observations available. Instead, we have compiled in Fig. 5 histograms that contain all observations carried out at altitudes between cloud boundaries. A similar procedure was used to build the histograms of the modeled vertical wind distributions at the same altitude of observations. We only show here histograms for the third and fifth hours of Case 1. Detailed information on specific sections inside the cloud can be found in Section 7 of the supporting information. For both hourly intervals, there is a high degree of correlation between model-based and radar-based distributions
of the vertical wind. As these distributions agree in terms of frequencies, variance, and skewness, average overlapping index values are above 0.77 for the selected hours. This significant degree of agreement between modeled vertical wind and observations repeats during the cloud event with average overlapping index values of $0.81 \pm 0.03$ for halo doppler lidar and $0.86 \pm 0.06$ for cloud radar. Comparing the panels of Fig. 5, we can confirm the increasing trend of turbulent intensity through the cloud domain. Particular trends in the turbulence dynamics can be observed at every cloud section, but in general, the turbulent
mixing decreases from the cloud base to the cloud top due to surface-driven conditions. The maximum updraft velocity goes from $1 \, \mathrm{m \, s^{-1}}$ to $1.6 \, \mathrm{m \, s^{-1}}$ between the third and fifth hours of the cloud event, and the $\sigma_w$ values of the distribution increase too. While turbulence-induced by cloud top radiative cooling weakens with time after sunrise, the surface-driven convection strengthens due to an increase in surface temperature.



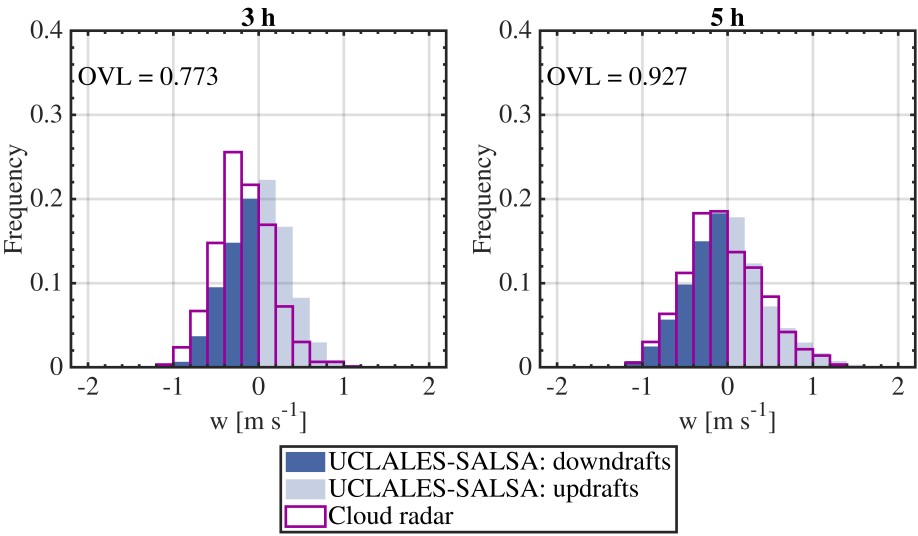

**Figure 5.** Comparison of model-based distributions of vertical wind at in-cloud conditions during Case 1 24 September 2020 to those retrieved from cloud radar observations. Each panel shows the overlapping index value (OVL) as an indicator of agreement between distributions.

### 3.1.3 Size dependent activation efficiency

We studied the cloud activation process by comparing the model-based and observation-based activation efficiency curves retrieved from aerosol particle number concentrations measured at the Puijo station with the Twin-inlet DMPS-system. Model-based number concentrations of activated droplets and activation efficiency curves were calculated following a size-based selection procedure that resembles experiments. We separated cloud droplets and aerosol particles with wet diameters below 40 µm to estimate total droplet number concentrations at 225 m, the Puijo station altitude. Likewise, we separated cloud droplets

and aerosol particles with wet diameters below 1 µm to calculate the number concentration of interstitial aerosol. This procedure was carried out in every grid point through the horizontal domain (225 m). Number concentrations of activated droplets and total aerosol were used to calculate the activated fraction per size bin. Activation efficiency curves were then obtained from horizontally averaged values in hourly intervals.

Figure 6 shows how the model follows nicely the shape of total and interstitial aerosol size distributions observed by the

Twin-inlet DMPS system using an aerosol sectional representation of 18 size bins. Modeled number concentrations of activated particles were later used to calculate the activated fraction per size bin together with the activation efficiency curves and values of the particle diameter for 50% activation efficiency or $D_{50}$ that are depicted in Figure 7. To assess the effect of turbulence fluctuations, we studied separately the activation efficiency in grid points with updraft winds or downdraft winds. More information about averaging and treatment of model outputs related to these calculations can be found in section 8 of the

supporting information.





During Case 1, observations indicated small changes in the curve slope and $D_{50}$ values likely because of the low variability in aerosol composition and number concentrations. Observed $D_{50}$ values not shown here, decrease monotonically from 0.188 μm to 0.156 μm between the first and fifth hour of the cloud event, respectively. The largest reduction occurred after the first hour when $D_{50}$ decreases to 0.167 μm (see Section 9 of the supporting information). This change is likely due to an increase in the

cloud base altitude, and moving from fog dynamics to cloud dynamics. Model-based and observation-based activation efficiency curves in Fig. 7 were in close agreement in terms of both, $D_{50}$ value and the slope of the sigmoidal section showing that the model captured well the dynamics of the droplet activation process. Since aerosol properties did not change significantly, the reduction of $D_{50}$ values indicated an enhancement in droplet activation promoted by larger surface heat fluxes and stronger turbulent circulation. Stronger and more variable updrafts also affect activation efficiency curves. Figure 7 shows that curves

calculated for updrafts and downdrafts became significantly different between them. At the fifth hour, aerosol particles with sizes below $D_{50}$ that are activated in updrafts might become non-activated in downdrafts. The difference between up- and downdrafts increases during the simulation as the cloud ascends and observation altitude moves closer to the cloud base.

We compared the average supersaturation for droplet activation in UCLALES-SALSA to the effective supersaturation $SS_{eff}$ for droplet activation at equilibrium conditions given by the $\kappa$-Köhler model (Petters and Kreidenweis, 2007). $SS_{eff}$ was cal-

culated using average $D_{50}$ values from observations and a volume-weighted average $\kappa$-value of 0.356 based on the observed aerosol composition. To calculate the average supersaturation at droplet activation in UCLALES-SALSA, we matched maximum supersaturation values ($SS_{max}$) to the cumulative number concentration of activated droplets ($N_{d,act}$) through vertical columns in those grid points of the model domain driven by updrafts. Hourly supersaturation values were calculated as averages weighted by $N_{d,act}$ number concentrations. Since the wet size of the largest interstitial aerosol particles modeled by

UCLALES-SALSA exceeds occasionally 1 μm in these specific conditions, instead of using the $D_{50}$ value retrieved from a cut off size of 1 μm, we calculated $SS_{eff}$ based on $D_{50}$ values obtained with a cut off size of 2 mum to differentiate better between interstitial aerosol particles and cloud droplets. More information about these calculations is included in Section 8 of the supporting information.

$D_{50}$ values reflect modeled cloud activation at Puijo altitude (225 m) located at cloud top height at the beginning of the

cloud event and later located at cloud base height at its end. We found that during the first hour, the $SS_{eff}$ for a modeled $D_{50}$ of 0.191 μm is equal to 0.081%, a value lower than the 0.107%, average-SS for droplet activation calculated in UCLALES-SALSA during the droplet activation. From the second hour, the analyzed $D_{50}$ from model data increased steadily from 0.174 μm to 0.196 μm in the fifth simulated hour, corresponding to a decrease in $SS_{eff}$ from 0.092 down to 0.077. At the same time, the average-SS during activation increased from 0.122 to 0.163 as the strength of modeled updrafts increased. This again

indicates that a large fraction of droplets evaporated inside the cloud after activation, producing a vertical profile with increasing average droplet number concentration as a function of altitude. Also, as the observed $D_{50}$ value leads to very low estimates of supersaturation at the cloud base during the activation, the employment of typical cloud droplet formation parameterizations based on an updraft velocity probability distribution, would have overestimated the average cloud droplet number concentration in the cloud.





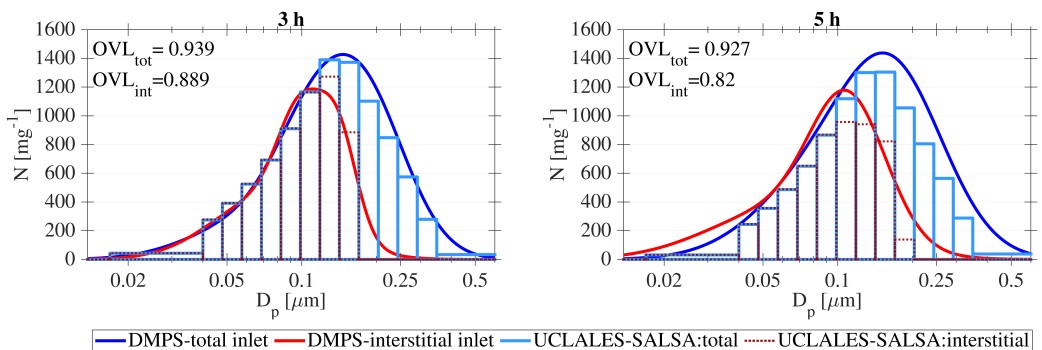

**Figure 6.** Comparison of aerosol size distributions calculated with UCLALES-SALSA at Puijo altitude of 225 m and measured the Twin-inlet DMPS system during Case 1 24 September 2020.

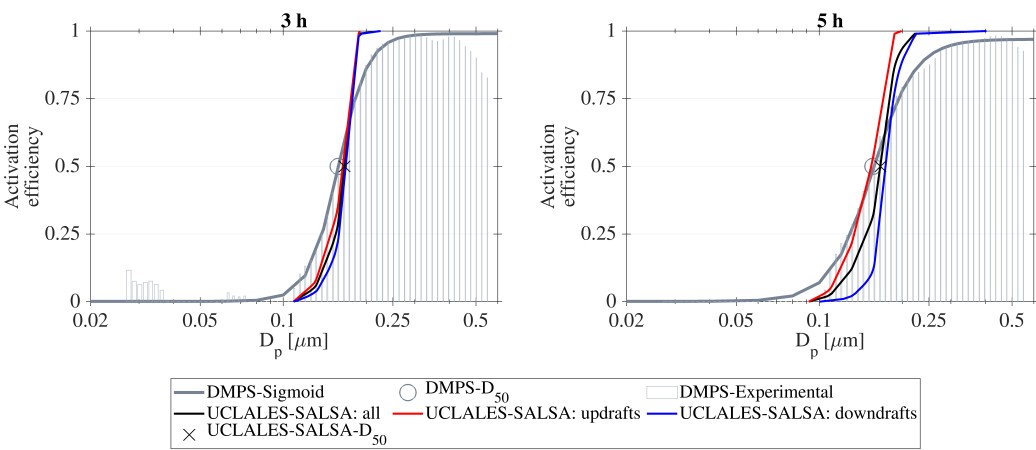

**Figure 7.** Comparison of activation efficiency curves calculated with UCLALES-SALSA at Puijo altitude of 225 m and retrieved from aerosol number concentrations measured by the Twin-inlet DMPS system during Case 1 24 September 2020.




Since Case 1 occurred during the biomass burning plume period, it is likely to have an externally mixed aerosol population composed of two types of particles, particles locally emitted or formed in situ, and particles from aged biomass burning emissions transported long range. Unfortunately, measurements do not provide information on the aerosol mixing state. Despite that, to assess the potential effect of the aerosol chemical diversity in our simulations, we compared the simulation results obtained for an internally mixed aerosol population with those for an externally mixed aerosol population with the same aerosol

number size distribution. As expected, the slopes in activation efficiency curves of the externally mixed aerosol population were less steep than those for the internally mixed aerosols and match better the observed slopes. Nevertheless, there were no significant changes in $D_{50}$ values nor in droplet number concentrations and size distributions. Detailed information is included in Section 9 of the supporting information.

### 3.1.4 Droplet microphysics

To assess the modeling closure for droplet microphysics, we compared model-based droplet size distributions to observations carried out by the FM-120 and the ICEMET instruments. A more detailed analysis of the sources of inter-instrument variability related to differences in time and bin size resolution, observational range, and sampling conditions was presented in Tiitta et al. (2022). Model-based size distributions for hydrometeors were obtained as horizontally averaged values for 1 h long intervals. Turbulent convective circulation through the model domain induces large variability in droplet microphysics, e.g. even at

the same time and altitude, dry particles with equivalent size and composition can show different wet sizes depending on water balance at the grid point. Since UCLALES-SALSA uses common bin microphysics based on dry particle size, before performing any averaging operation, it was necessary to group hydrometeors into size-resolving microphysics based on wet size. Consecutive size bins for wet size have a volume ratio of 2-3 with values ranging from $0.5\,\mu m$ to 2 mm.

    Figure 8 compares hourly average observed size distributions with model results for total droplet concentrations including

cloud and drizzle droplets. Droplet distributions from the different instruments correlate to each other where observational ranges overlap. Results of an intercomparison study on the performance of both instruments during the Puijo 2020 sampling campaign are provided in (Tiitta et al., 2022) and especially the sampling of larger droplets was found to be highly sensitive for the wind direction. The shape of droplet size distributions follows the observations closely over the measured size range, again demonstrating the skill of the model to reproduce the growth of cloud droplets. Since droplet formation evolves under

a constantly high aerosol loading (c.a. $1000\,cm^{-3}$) of moderate hygroscopicity and a median size of $0.2\,\mu m$, the droplet size distribution at the early stage of cloud formation is narrow with a mean droplet size of $10\,\mu m$. Collisional droplet growth is limited since collision efficiency for droplet pairs with sizes ranging between $1\,\mu m$ and $10\,\mu m$ is very low compared to that observed for large droplet pairs (e.g. $10\,\mu m$ and $20\,\mu m$) (Pinsky et al., 2008). In Fig. 8 we notice how the droplet size distribution shifted towards smaller sizes and number concentrations increased c.a. 50% for droplet sizes below $6\,\mu m$ between

the third hour and the fifth hour. Under increasing strength and variability of updrafts, the constant formation of new activated droplets leads to a droplet size distribution dominated by smaller droplets with low collisional growth rates and curvature-enhanced evaporation. Also with stronger turbulent mixing, the residence time is shorter limiting the condensational growth of larger aerosol particles and larger droplets as well as their number of collisions. Both effects translate into a reduction of the





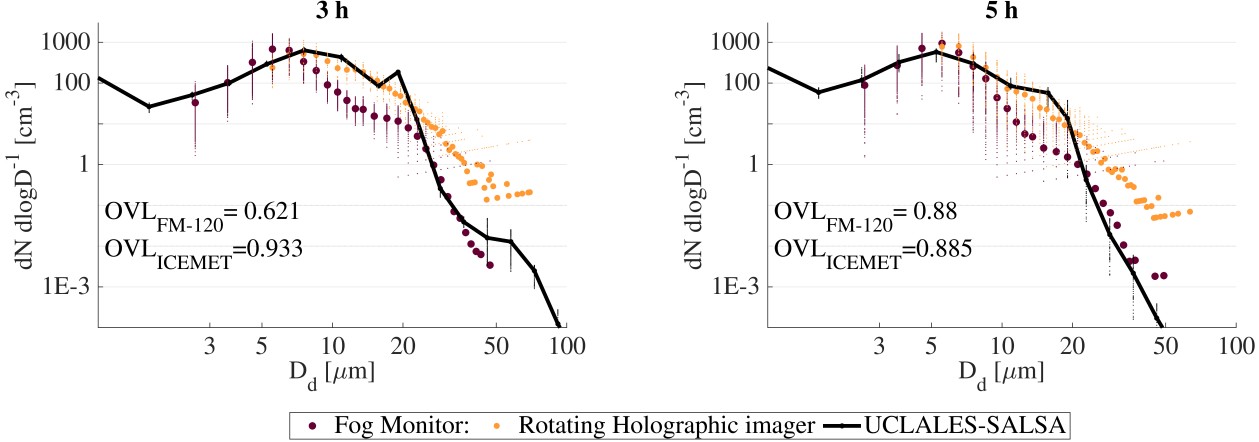

**Figure 8.** Model outputs of droplet size distributions at Puijo altitude of 225 m for Case 1 September, 24[th], 2020 compared to observations from the fog monitor (FM-120) and the holographic imaging system (ICEMET). Overlapping index values (OVL) are included as indicators of agreement between distributions.

right tail of the droplet size distribution with the consequence suppression of drizzle formation. Droplet size distributions and
overlapping index values for both simulation scenarios (i.e. internally mixed and externally mixed aerosols) are included in Section 10 of the supporting information. Simulation outputs for every scenario are provided in the data repository (Calderón et al., 2022).

### 3.2 Case 2: Nocturnal cloud of 31 October 2020

### 3.3 Cloud boundaries

Unlike in Case 1, observation- and model-based cloud boundaries for the nocturnal cloud on 31 October 2020 change only slightly with time so that the liquid water content increases in the upper section of the cloud as reflected by Fig. 9a). Model results in Fig. 9b) show a very well-mixed stratocumulus capped boundary layer with cloud droplet number concentrations that do not vary significantly in the vertical direction. Although the modeled liquid water content profile follows the cloud development perfectly, modeled droplet number concentrations are different from observations. Causes of model biases are
explored later in the sensitivity analysis for this case. Droplet concentrations are on average one fourth of those observed for Case 1 as a consequence of the lower aerosol loading. In the absence of incoming solar radiation, the radiative cooling at the cloud top dominates the turbulence formation. In contrast with the diurnal cloud, the cloud-top cooling rate during Case 2 does not show any particular trend with respect to time. It varies between $83\ \mathrm{W\,m^{-2}}$ and $97\ \mathrm{W\,m^{-2}}$ with a mean value of $89.7\ \mathrm{W\,m^{-2}} \pm 2.2\ \mathrm{W\,m^{-2}}$ (see section 6 of the supporting information).



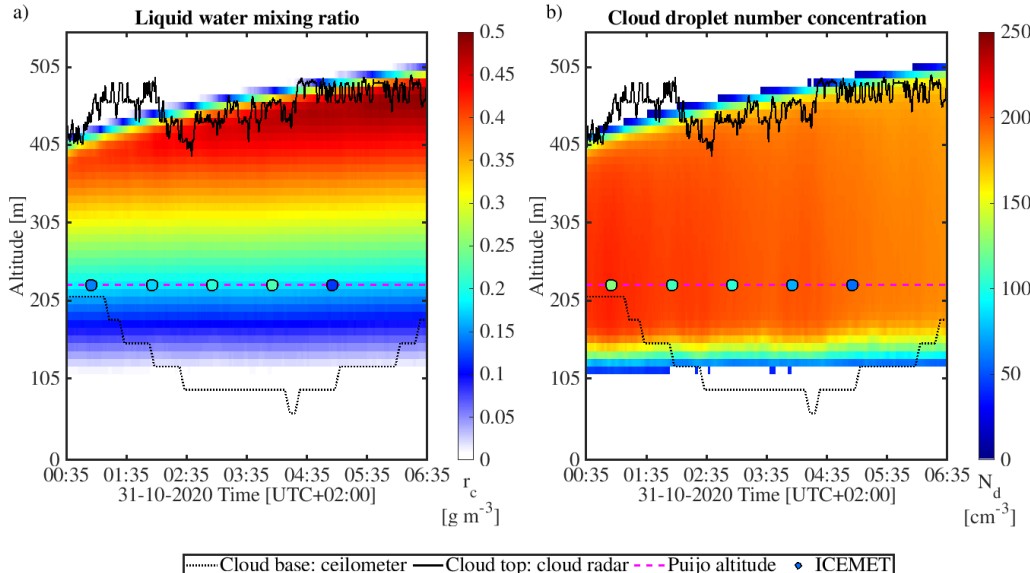

**Figure 9.** Comparison of cloud boundaries for Case 2 31 October 2020 defined by modeled liquid water content and cloud boundaries retrieved from cloud radar and ceilometer observations and a) modeled vertical profile of liquid water content and observations from the holographic imaging system (ICEMET) b) modeled cloud droplet number concentrations and observations from the holographic imaging system (ICEMET).

### 3.3.1 In-cloud vertical wind distribution

There is a good agreement between distributions of modeled and observed vertical wind velocities at the cloud base. The turbulence was stronger compared to the diurnal event (Case 1) but did not change significantly with time. According to the model at cloud base, the updraft velocity standard deviation varies between $0.4\,\mathrm{m\,s^{-1}}$ and $0.5\,\mathrm{m\,s^{-1}}$ with maximum values of updraft velocity around $1\,\mathrm{m\,s^{-1}}$. In Fig. 10 we notice that the model tends to overestimate the frequency of strong downdrafts during the first hour. At the beginning of the cloud event, the surface is warmer than the air in contact with it and adds moisture and energy to the boundary layer during its cool down. If these surface fluxes are being underestimated by the model, negative buoyant fluxes associated to cloud-top radiative cooling effect, could be positively biased. Nevertheless, these biases are not significant for the remaining hourly intervals, and the model represents well the distribution of updrafts/downdrafts at the cloud base. Corresponding histograms are included in Section 7 of the supporting information.

With respect to the vertical wind distribution in other cloud sections, we found that model-based distributions of the vertical wind agree reasonably well with radar observations in terms of frequency, variance, and skewness at all altitudes, but just until the end of the second hour. After this time, retrieved distributions of vertical velocity are shifted towards negative velocities indicating the formation of drizzle and also ice particles at the upper section of the cloud. We can see in Fig. 11 how this phenomenon affects the average distribution of vertical wind in the cloud domain at the sixth hour. During drizzle or snow,



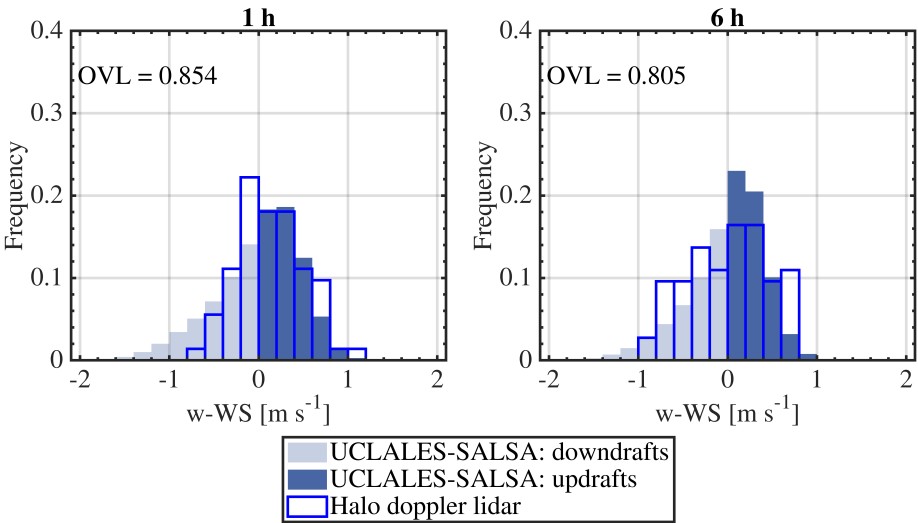

**Figure 10.** Comparison of model-based distributions of vertical wind at cloud base during Case 2 31 October 2020 to those retrieved from Doppler lidar observations. Each panel shows the overlapping index value (OVL) as an indicator of agreement between distributions.

the cloud radar signal is mainly dominated by larger falling hydrometeors becoming blind to small droplets carried up during updrafts (Bühl et al., 2015), therefore, the velocity profile retrieved from the cloud radar cannot be used as a proxy for the vertical velocity of air similar to Case 1. This explains why calculated and observed histograms do not match as they did previously. The model output for vertical wind includes just the turbulent air velocity and it is not affected in any form by the sedimentation velocity of hydrometeors. In order to compare against radar retrieval, we must emulate the observed radar Doppler spectra using model outputs for vertical wind, wet size, and number concentrations of all hydrometeors in the cloud domain. Results for this part of the closure study are explained later in this section as they are highly dependent on droplet microphysics.

### 3.3.2 Size dependent activation efficiency

During Case 2, a fast reduction in aerosol number concentrations from an initial value of $200\,\mathrm{cm}^{-3}$ to $76\,\mathrm{cm}^{-3}$ in the accumulation mode was observed. This was accompanied by a high variability in the aerosol composition, thus affecting the ability of the model to represent the change in conditions and droplet number concentration. This can be seen in Fig. 12, where observed and modeled number concentrations of total and interstitial aerosol are compared. Although the model follows the shape of aerosol size distributions over time, it cannot fully describe the particle behavior in both, the Aitken and accumulation modes. Nevertheless, these biases have a moderate effect on the closure between model-based and observation-based activation efficiency curves that are shown in Fig. 13. For the first simulated hour, the model reproduces the dry particle size of mean



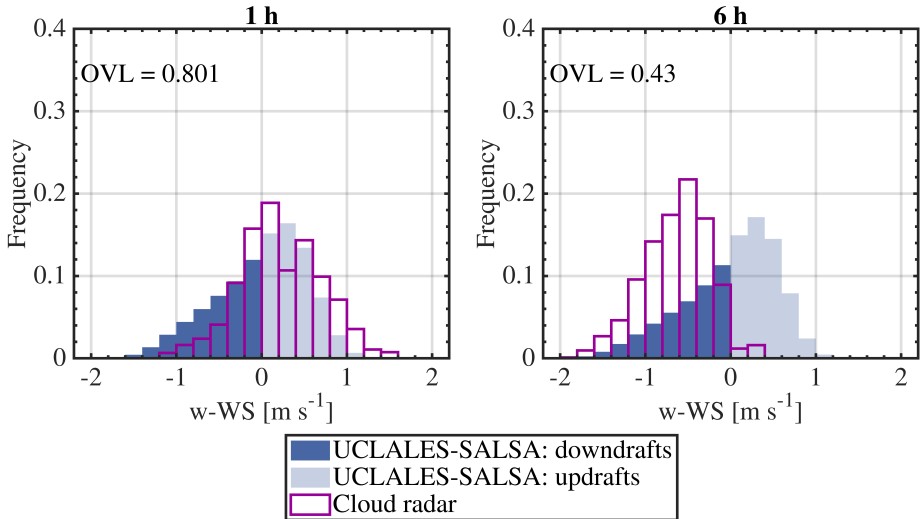

**Figure 11.** Comparison of model-based distributions of vertical wind at in-cloud conditions during Case 2 31 October 2020 to those retrieved from cloud radar observations. Each panel shows the overlapping index value (OVL) as an indicator of agreement between distributions.

activation or $D_{50}$, but the slope differs. These biases in model predictions can be attributed to changes in ambient aerosol composition and aerosol number concentrations caused by changes in air mass origin that are not accounted for in our simulations. They can also be related to the uncertainty in observations related to low and varying aerosol content as was suggested by the positive difference between interstitial and total aerosol particles below 30 nm that is shown in Fig. 12, as well as by the

apparent activation of aerosol particles as small as 30 nm that can be seen in Fig. 13. Droplet activation at this dry particle size is not realistic in these conditions (i.e. droplet activation would require a supersaturation of 1.8% according to the $\kappa$-Köhler theory, a value well above the maximum supersaturation in strongest updrafts). AMS-measurements also indicated variable aerosol composition over hourly intervals of Case 2. On the contrary to what was observed during Case 1, observation-based curves show more variability and lower $D_{50}$ values in the nocturnal cloud as seen in Fig. 13. During the first three hours, curves

progressively become less steep and the $D_{50}$ value show a positive trend. After the fourth hour, these trends reverse and curves become steeper with smaller $D_{50}$ values ranging between $0.092\,\mu m$ and $0.094\,\mu m$. These changes in the shape of activation curves correlate well with changes in AMS-based aerosol composition from organic-enriched to more inorganic aerosol particles. These changes in the slope of observed activation efficiency curves suggest that aerosols evolve from an externally mixed population to a more internal one with homogeneous composition. Less steep curves where the activation efficiency increases

slowly with increasing size have been observed in externally mixed aerosol populations (Anttila, 2010; Deng et al., 2011; Väisänen et al., 2016; Vu et al., 2019).

In Case 2, the effective supersaturation $SS_{eff}$ calculated from aerosol composition (volume-weighted average $\kappa$ value equal to 0.237) and modeled-average $D_{50}$ is 0.287% $\pm0.004$, a value that approaches well the average SS for droplet activation of





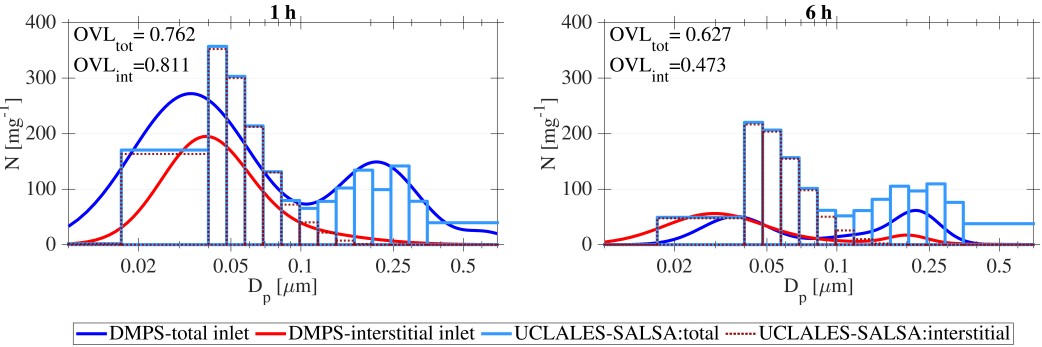

**Figure 12.** Comparison of aerosol size distributions calculated with UCLALES-SALSA at Puijo altitude of 225 m and measured by the Twin-inlet DMPS system during Case 2 31 October 2020

0.289 % ±0.006 obtained with UCLALES-SALSA. This similarity suggests that droplet evaporation is not as important as it
was for Case 1. Since there were no significant changes in the modeled distribution of in-cloud supersaturation values during
the cloud event, biases in modeled activation efficiency curves are likely related to changes in aerosol composition (i.e. gas-to-particle conversion of gaseous sulfur emissions) or number concentrations that were not accounted for in our simulations
(i.e. mixing with different air mass during horizontal entrainment). For Case 2, we initialized our simulation with dry aerosol
particles in a single mixing state composed of 88 % v/v of organic carbon and 12% v/v of sulfate. This composition was
estimated from average values of ACSM measurements and AMS measurements in the hourly interval prior to the cloud event.
A better agreement between model-based and observation-based curves for the first hour suggests that our settings for the
aerosol composition could have been more representative of aerosol size-dependent hygroscopicity at the beginning of the
cloud lifetime. During this event, the wind direction varied thoroughly in the range between 128 degrees and 360 degrees
through a wide sector with several local aerosol sources (i.e. heating plant, highway, residential areas) rising the probability
of having variability in the atmospheric background, different from the one used to initialize our simulation. Unfortunately,
detailed information on aerosol composition is not available due to limitations in the time resolution and accuracy of aerosol
observations.

### 3.3.3   Droplet microphysics

Opposite to Case 1, UCLALES-SALSA predicts a well-mixed boundary layer with total droplet number concentrations that
do not vary significantly with increasing altitude but decrease from $210\,\mathrm{cm}^{-3}$ to $180\,\mathrm{cm}^{-3}$ between the beginning and the
end of the cloud event. In terms of droplet size, although we were lacking direct observations of large and drizzling cloud
droplets, the shape of droplet size distribution follows the observations over the measured size range as shown in Fig. 14.
Like in the case of activation curve, also the narrower modeled droplet size distribution can be attributed at least partly to the
lack of variability in aerosol properties in the modelling results. In an additional step to assess the modeled droplet spectrum



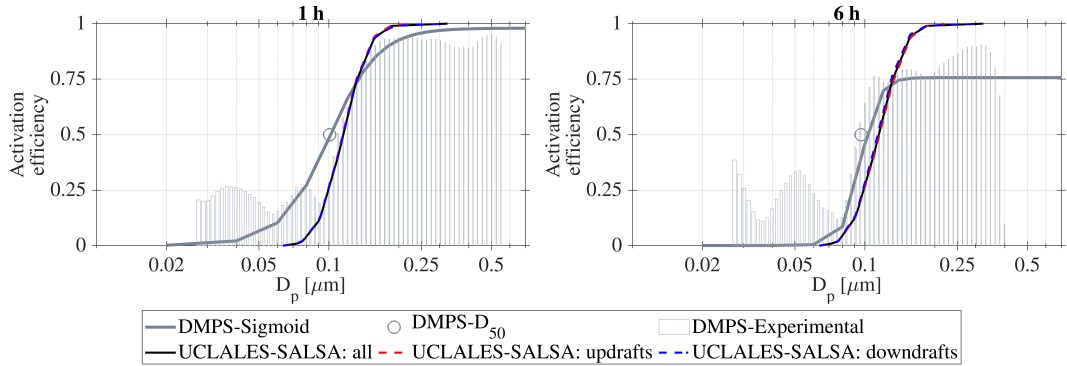

**Figure 13.** Comparison of activation efficiency curves calculated with UCLALES-SALSA at Puijo altitude of 225 m to those retrieved from aerosol number concentrations measured by the Twin-inlet DMPS system during Case 2 31 October 2020

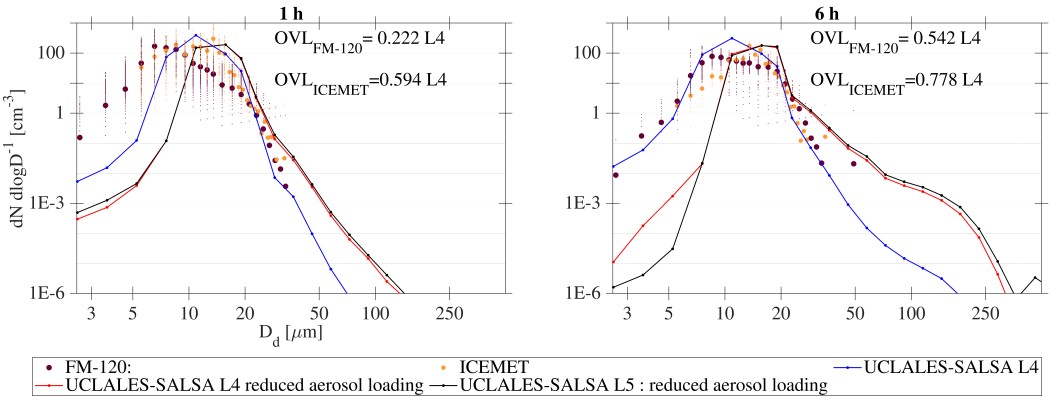

**Figure 14.** Modeled droplet size distributions at Puijo altitude of 225 m for Case 2 31 October 2020 compared to observations from the fog monitor and the holographic imaging system. Overlapping index values (OVL) are included as indicators of agreement between distributions.

for our base simulation, we calculated the settling velocity of all hydrometeors in the cloud domain using model outputs for size-segregated number concentrations. Then, we estimated the average settling velocity of the droplet spectrum and added it to the vertical wind velocity at each grid point of the model domain to emulate observer radar Doppler spectra at Puijo altitude. Despite the moderate agreement between model-based and observation-based droplet size distributions, settling velocities and droplet sizes did not replicate the observed distribution of radar velocities inside the cloud. Details of these calculations are included in section 11 of the supporting information.

### 3.3.4  Model sensitivity analysis to inputs of aerosol number concentrations in simulations of Case 2

As model biases in aerosol number concentrations and activation efficiency curves pointed out the aerosol properties as the most likely cause of discrepancy from observations of cloud activation efficiency, we decided to investigate up to what extent



the modelling results are dependent on aerosol composition and number concentration. To do so, we performed two additional simulations with identical initial atmospheric thermodynamic state, time, and domain resolution but changed the aerosol properties used for initialization. In the first scenario, we kept the same aerosol composition and shape of the number size distribution, but reduced the total aerosol number concentration used at the model initialization by 40%.

With a lower aerosol loading, the model predicts a significant reduction in the mean activation efficiency diameter (see section 8 of the supporting information). Horizontally averaged total droplet number concentrations drop proportionally to the aerosol number concentration showing decreasing average values between $134\,\mathrm{cm}^{-3}$ to $117\,\mathrm{cm}^{-3}$, which are only 64% and 63% of those calculated in the base simulation scenario.

Therefore, larger droplets are produced with the same liquid water content, as the last depends just on the thermodynamic state of the atmosphere. Now, droplet settling velocities displace our estimated distributions of radar velocity to the left as expected. The improved agreement between modeled and observed distributions of the vertical wind through the cloud domain is indicated by overlapping index values that vary between 0.778 and 0.94 as can be seen in Fig. 15.

During the third hour, once a significant fraction of aerosol particles have activated to cloud droplets and the aerosol loading has decreased significantly due to both in-cloud activation and scavenging, large droplets that have grown to reach diameters above $50\,\mu\mathrm{m}$ cause a broadening of the droplet spectrum due to their larger settling velocity (e.g. settling velocity of droplets increase by 2 orders of magnitude when droplets grow from $6\,\mu\mathrm{m}$ to $60\,\mu\mathrm{m}$ in diameter). The emulated velocity spectra overlap with observations showing a long tail to the left towards stronger downdraft velocities as shown in Fig. 15. During the next hours, this broadening of the radar velocity distribution proceeds slowly because the majority of droplets still have sizes around $10\,\mu\mathrm{m}$ to $20\,\mu\mathrm{m}$ with low collision efficiency values (Chen et al., 2018b). In time, the radar velocity distribution becomes more skewed to the left because cloud processing is producing larger aerosol particles, which turn into drizzle-sized droplets faster in the cloud domain due to collisions with smaller droplets. Observations at Puijo altitude reported in Fig. 14 confirmed this trend as droplet number concentrations for droplets with sizes above $50\,\mu\mathrm{m}$ increase while there is a persistent fraction of cloud droplets with sizes between $10\,\mu\mathrm{m}$ and $20\,\mu\mathrm{m}$ that moves slowly to smaller droplet sizes during the cloud event. In the sixth hour, the broadening of the droplet size distribution has produced a wide range of settling velocities that go from low values in the order of $0.003\,\mathrm{m\,s}^{-1}$ corresponding to cloud droplets to large drizzle droplets reaching a maximum falling rate of $1.05\,\mathrm{m\,s}^{-1}$, equivalent to the terminal velocity of a raindrop with $2.5\,\mathrm{mm}$–$2.7\,\mathrm{mm}$ in diameter. Nevertheless, there are negative biases in the left branch of the emulated radar velocity which indicate that the fraction of large droplets in the droplet population is not enough to replicate the velocity values observed by the cloud radar.

Based on the relevant role of droplet size in the degree of modelling closure for Case 2, we investigate the effect of ice formation as this process can produce larger hydrometeors that can displace further the left tail of the radar velocity distribution. Both, model outputs and observations, showed subzero temperatures of approximately -4°C at the upper section of the cloud. Light snow was confirmed visually and via weather sensor, and the depolarization signal from cloud radar confirms the formation of mixed phase cloud with low ice content (Li et al., 2021). To gain insights into this, we performed an additional simulation with a reduced aerosol loading and the same initial profile of the atmospheric thermodynamic state, time, and spatial





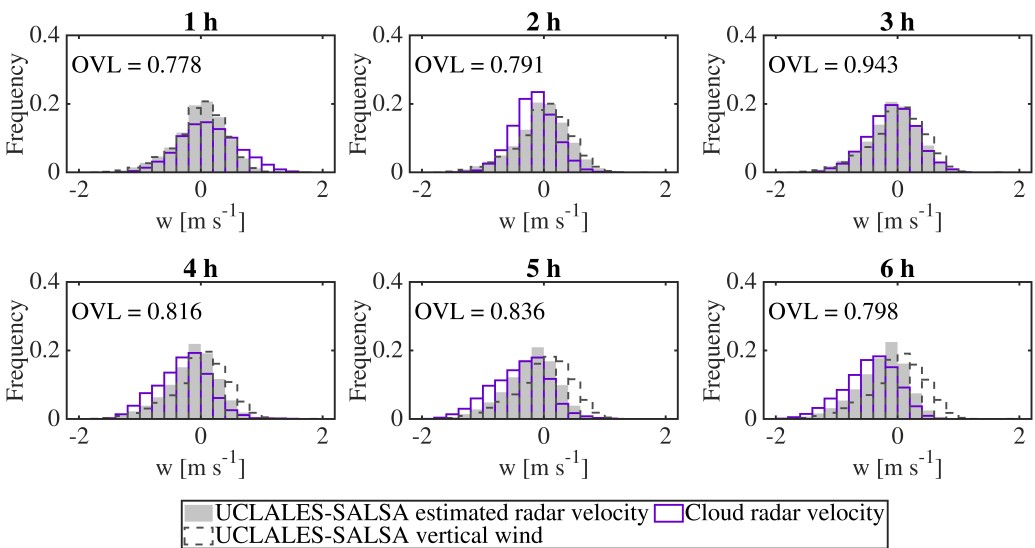

**Figure 15.** Comparison of distributions of radar velocity retrieved from cloud radar observations at in-cloud conditions to those calculated by UCLALES-SALSA for Case 2 of 31 October 2020 in the simulation scenario with 40% reduction in aerosol loading used for model initialization without ice formation. Overlapping index values (OVL) are included as indicators of agreement between distributions.

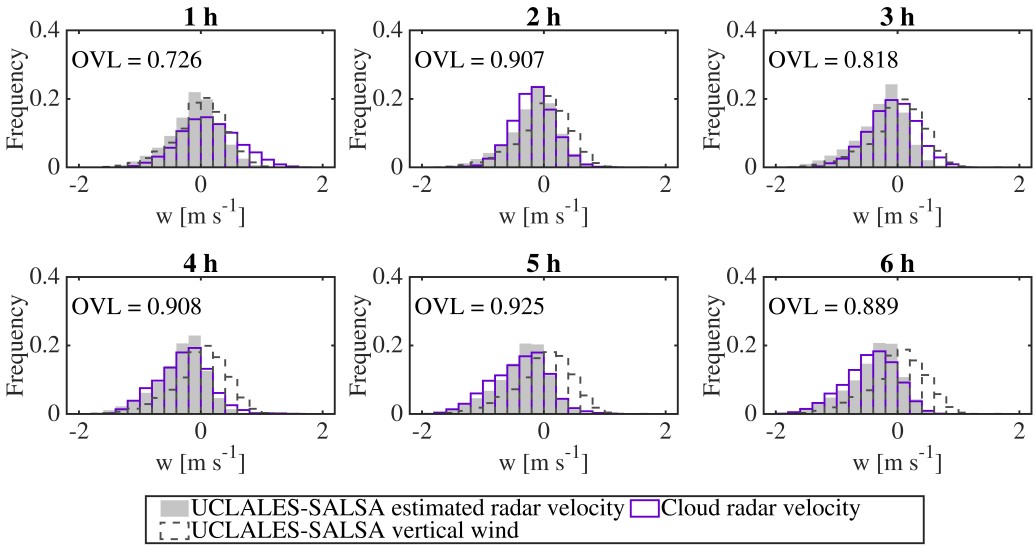

**Figure 16.** Comparison of distributions of radar velocity retrieved from cloud radar observations at in-cloud conditions to those calculated by UCLALES-SALSA for Case 2 of 31 October 2020 in the simulation scenario with 40% reduction in aerosol loading used for model initialization with ice formation. Overlapping index values (OVL) are included as indicators of agreement between distributions.





resolution of the model domain, but this time turning on the module for calculating ice formation and ice-related processes. At this temperature, ice formation must occur heterogeneously with the help of ice-nucleating particles mainly via immersion.

For this simulation scenario, we kept the shape parameters of the aerosol size distribution but divided the aerosol particles in two mixing states, 85% of the total aerosol number concentration kept the same composition (88%v/v organic carbon. 12% v/v sulfate), and the remaining 15% was assumed to be composed of dust and sulfate with volume fractions of 90.5% and 9.5%, respectively.

Unlike in the scenario of reduced aerosol loading, there were no significant differences between activation efficiency curves

calculated with and without ice formation (see section 8 of the supporting information) but drizzle microphysics was displaced toward larger sizes with higher number concentrations. Ice particles were formed from the beginning with increasing size and number concentrations during the cloud event but within orders of magnitude below detection limits of the fog monitor and the holographic imaging system. Without observations to validate model outputs of droplet size distributions at the expected size range of ice particles, we took advantage of the distribution of radar velocities observed during the cloud event to perform

a comparison with the velocity distributions derived from vertical wind velocity and droplet size and number concentrations obtained with UCLALES-SALSA. A description of these calculations is included in Section 11 of the supporting information.

Even when simulated ice number concentrations are low, below $7\,\mathrm{m}^{-3}$ at Puijo level, including ice processes in our simulation further improved the modeled radar velocity distribution as compared with observations during the last three hours of the event with overlapping index values above 0.89 as seen in Fig. 16. In the second hour, droplet settling velocities range between

$0.03\,\mathrm{m\,s}^{-1}$ and $0.4\,\mathrm{m\,s}^{-1}$ and the emulated radar velocity agrees better with the observed one in comparison with simulations that were performed without ice effects. This rapid size transition toward larger sizes is caused by the rapid activation and growth of the ice-nucleating particles used to initialize the simulation. These large particles with a dry diameter around 0.7 μm become large droplets that are less susceptible to evaporation shrinkage in downdrafts and grow efficiently by collection of smaller droplets. At the third hour, collisional growth of larger hydrometeors, including ice particles, has proceeded quickly and

the overlapping index drops from 0.943 to 0.818. However, with larger droplet sizes, collisional growth is also more efficient and become enhanced by turbulence fluctuations (Yang et al., 2018; Chen et al., 2018b) and positive buoyant fluxes locally induced by droplet sedimentation (Mellado, 2017). Thus, at the fourth hour, the broadening of the droplet size distribution accelerates the growth of drizzle and ice particles, and the closure for radar velocity distributions is greatly improved. Without detailed observations of ice particles, it is difficult to validate the model assumptions for ice processes (e.g. total ice mass, ice

shape, ice density), however, the contrasting differences between our simulations of Case 2 prove that perturbations in aerosol properties can have a profound effect on cloud microphysics if aerosol loading is low. Even with a moderate degree of modelling closure for vertical wind distribution and CCN concentrations, we might fail in our estimations of cloud properties based on droplet microphysics such as the effective droplet diameter or the median volume diameters, common proxies for cloud optical depth and liquid water path. Model outputs for all simulation scenarios are provided in the data repository (Calderón

et al., 2022).





## 4  Conclusions

We have used UCLALES-SALSA to study changes induced by atmospheric dynamics and aerosol-cloud interactions in stratocumulus clouds formed in a boreal environment with anthropogenic influence. The use of in situ and remote sensing observations to initialize the atmospheric thermodynamic state and aerosol properties was essential for the successful simulation of
cloud properties. Observed aerosol size distribution and chemical composition proved to be representative of cloud base CCN, as the model could follow the droplet activation process as well as the time evolution of aerosol and hydrometeor microphysics. We found a significant effect of the vertical wind intensity and variability on cloud droplet size distribution and number concentrations (CDNC). It is also presented how closure studies can be extended from aerosol-droplet concentration comparison to include aerosol size-dependent properties, boundary layer dynamics, and droplet size distribution with novel modelling tools
and comprehensive observations.

   In the first case study, cloud formation occurred in relatively polluted atmospheric conditions during the daytime, where increasing strength of the boundary layer mixing induced by surface-driven turbulent mixing caused significant differences in droplet average number concentration between cloud base and cloud top. High aerosol loading decreased the mean droplet size, leading to fast evaporation of droplets in the downdrafts, thus producing high variability in the cloud droplet concentration in
the lower part of the cloud. Such variability should be accounted for when analyzing the in situ observations as the measured susceptibility of droplet number concentration on changes in aerosol seemingly depends on the relative altitude of observations inside the cloud.

   In the second case study, the cloud formation occurred in clean atmospheric conditions during nighttime, with boundary layer circulation driven by radiative cooling from the cloud top. The temperature was also low enough to allow a formation
of a small amount of ice during the event. Low aerosol loading allowed activation of smaller aerosol particles due to higher supersaturation values compared to those observed in polluted conditions. Beyond, the presence of large aerosol particles in the accumulation mode favored the rapid formation of wide droplet size distribution where large droplets grew effectively through turbulence-enhanced collision-coalescence to produce drizzling droplets. Opposite to the first cloud case, the droplet number concentrations did not show vertical variability but changed rapidly in time.

Observations, such as those conducted in the Puijo tower, provide information on size dependent activation of aerosol particles, and this information has also potential to shed light on relevant cloud processes, such as entrainment mixing or in-cloud evaporation. However, to gain more information based on observed activation efficiency curves, more detailed information on aerosol size-dependent hygroscopicity is needed (Case 1), and also the variability of aerosol particle properties (Case 2). In case of low aerosol loading, the current observational methods have too high uncertainty, and thus the possibility to constrain
detailed model processes based on observations is limited.

   We highlight the importance of collecting more observations of in-cloud properties as they can decrease the uncertainty related to hydrometeor aggregation processes, especially those involving ice particles. It is important to reduce the gap of knowledge about the ice-nucleating ability of aerosol particles of both, natural and anthropogenic origin, as mixed-phase clouds have very different dynamics and radiative properties compared to liquid clouds.





Beyond providing information on the detailed microphysical processes taking place in the clouds, this study provides data that model developers can use to validate their models and to conduct sensitivity studies. To this end, the models have been quite commonly compared against observations from DYCOMS II, RICO, or MPACE, to mention a few, that usually are based on the airborne observations over a relatively short period of time. This study provides two well-characterized cloud events that can be used by the cloud modelling community to test their model frameworks for the aerosol-cloud droplet-precipitation-
turbulence interactions.

As the effect of cloud processing on aerosol properties is difficult to constrain using observations, the findings support the further employment of models like UCLALES-SALSA for the development of wet scavenging schemes accounting for different chemical compounds in global models.

*Code availability.*    Large-eddy-simulations were performed with UCLALES-SALSA (DEV version 17.08.2021) available from
https://github.com/UCLALES-SALSA/UCLALES-SALSA/tree/DEV. Input files and simulation outputs used in this research are available from https://fmi.b2share.csc.fi/records/81a8f2f7c854465cb6b362cfdc8f19c4 (Calderón et al., 2022)

*Data availability.*    The ground-based remote-sensing data used in this article was generated by the European Research Infrastructure for the observation of Aerosol, Clouds and Trace Gases (ACTRIS) and are available from the ACTRIS Data Centre using the following link: https://hdl.handle.net/21.12132/2.ef1a7d312c8a402d.

*Supplement.* There is supplementary information available for this study including:

1.  UCLALES-SALSA modelling framework
2.  Instrumentation used during the Puijo 2020 campaign
3.  Description of cloud cases
4.  Aerosol properties
5.  Variability of cloud properties and cloud radar observations
6.  Temperature and net radiative flux profiles
7.  Vertical wind distributions
8.  Cloud droplet activation and activation efficiency curves
9.  Model sensitivity analysis to inputs related to aerosol mixing state in simulations of Case 1
10.  Cloud microphysics and derived quantities
11.  Emulation of the radar Doppler velocity

*Author contributions.*    SC, JT, SR, HK and AV planned the study. AB, JT, MK, AL, HL, DM, IP and PT performed the observations and provided the data. SC made the model simulations. SC, JT, HK and SR performed the analysis, and SC prepared the manuscript with contributions from all co-authors.



*Competing interests.* TEXT

The authors declare that they have no conflict of interest.

*Acknowledgements.* Authors thank the Copernicus Climate Change Service (C3S) Climate Data Store for the ECMWF-ERA5 data down-loaded from https://www.ecmwf.int/en/forecasts/datasets/reanalysis-datasets/era5 (Hersbach et al., 2020), also thank the Aerosol, Clouds and Trace Gases Research Infrastructure (ACTRIS) for providing the CLU (2022) dataset in this study (CLU, 2022). CLU data was produced by

the Finnish Meteorological Institute (FMI) and is available for download from https://cloudnet.fmi.fi/.
This research has been supported by the Academy of Finland (grant nos. 309127, 317373), and the Horizon 2020 Research and Innovation Programme (grant no. 821205).



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
