# Peer review of "Aerosol-stratocumulus interactions: Towards a better process understanding using closures between observations and large eddy simulations"

_Atmospheric Chemistry and Physics, 2022_

## Referee Comment (RC1)

Review report of manuscript "Aerosol-stratocumulus interactions: Towards a better process understanding using closures between observations and large eddy simulations" by Calderon et al.,

**General comment**

The authors conducted a closure study to evaluate the UCLALES-SALSA with in situ station measurements during Puijo 2020 campaign. The authors show two cases with different aerosol loadings and meteorology (diurnal v.s. nocturnal) with the boundary layer profiles, cloud macrophysics, and microphysics properties. Sensitivity studies for the second case show that ice processes lead to better agreement between simulated radar velocity and the observed one. The authors highlighted the importance of more observations involving ice particles to reduce the gap in knowledge about the ice-nucleating ability of aerosol particles of both, natural and anthropogenic origin. Overall, I think this is a great manuscript in the closure study of aerosol-cloud interactions. I suggest publishing after the authors address both my key and minor comments below.

**Key comments:**

1. I think section 3.3.4 is a very important part of this manuscript, which provides information on the closure of the cloud base velocity. Comparing Figure 11b, Figure 15f, and Figure 16f leads to the conclusion that both reducing the total aerosol loadings and turning on the ice processes leads to better agreements between the model and observations, but the hypothesis cannot be tested due to the lack of observations of ice/mixed phase hydrometeor. Please correct me if my understanding is not what the authors would like to convey. Please confirm Figure 11b is used to compare with Figure 15f and Figure 16f. If I am right, my question is whether the role of ice phase particles is only significant with low aerosol loading? What will the result look like if not reducing the aerosol loading? Also in the text, I don't find any reason to explain the choice of how much reduction in aerosol loading. How do you come up with the number 40% in line 582? Is there any support for that from either the observation side or previous studies?

2. In lines 146-149, the authors said, "the effect of local topography on observed cloud properties is limited to certain high wind conditions". What is the definition of a high wind condition? Do the two selected cases associate with high wind or low wind? The second case has a lower boundary horizontal wind than the first case. What are the roles of the topography in the clouds examined in these two cases?

3. Figure 1, case 1 is initialized with the boundary layer temperature close to measurements, but the initial temperature above 200m is lower than the observation. Why not use the observed profile to initialize the model?

4. In both cases, simulated activation efficiency has a steeper slope than the DMPS observed. The authors stated in lines 452-456, that it is due to the aerosol mixing state. Can you elaborate on that? My understanding is external mixing leads to a higher

insoluble fraction of aerosols so that the activation of particles is suppressed. But why large particles are suppressed more than smaller particles? Is external mixing on large particles expected more frequent than on smaller particles? I think a more detailed explanation is needed and literature is needed to explain this, although there is no observation to provide the mixing state of aerosols.

5. In Figure 8, the Rotating holographic imager shows a higher concentration on the large particle, and the fog monitor shows a smaller concentration between about 6-25 microns. These differences are not changed at both 3hrs and 5hrs. As a closure study, what is the reason for the differences? Model uncertainties or observation uncertainties?

**Minor comments:**

1. Figure 4 and Figure 10, change the color of the dark blue bar, otherwise, it is hard to tell the differences between UCLALES-SALSA updrafts with Halo Doppler lidar. Figure 5 and Figure 11 are good examples.

2. Line 436, change mum to $\mu m$.

3. Line 472, correct the format of the citation.

---

## Referee Comment (RC2)

Review of Calderon et al., acp-2022-373

SUMMARY

This study describes observations and simulations of aerosol-cloud interactions via two case studies of low stratiform cloudiness at a continental site in eastern Finland. The cases are characterized by a unique suite of measurements similar to a US DOE Atmospheric Radiation Measurement program site including in situ measurements of basic meteorological variables (pressure, temperature, horizontal winds, water vapor amount) as well as aerosol and cloud properties; and remote sensing of cloud and boundary layer dynamical properties (i.e., vertical velocity) from a W-band Doppler radar-radiometer system, ceilometer and Doppler lidar. The simulations are designed to simulate aerosol-cloud interactions in a highly detailed manner using large eddy simulations coupled to a size-resolved Eulerian aerosol/cloud microphysics framework. The overall goal of the study is to demonstrate the significant degree of microphysical closure that can be obtained with the modeling framework.

A vast (and impressive) amount of observational information was synthesized to provide a comprehensive reference for the simulations and while the purpose of the study was to demonstrate the performance of the model, there was relatively little information given on the model construction and configuration. What model information was given was rather spread out over the manuscript, which made it difficult to understand exactly how the simulations were run. In addition, several major points were made in the conclusions/discussion regarding microphysical mechanisms that were barely mentioned in the results and relevant figures – this felt rather incongruous. As a result, I recommend the manuscript be returned to the authors for major revisions.

MAJOR COMMENTS

- The study builds upon a large body of previous and contemporaneous work to define and characterize the case studies and describe the modeling framework. In general, I found that the authors relied so heavily on references to other works that I was not able to understand much about their approach without close reading of these other manuscripts. I feel strongly that the paper should include further relevant details such that it can stand on its own. This is particularly the case with respect to the microphysics scheme, for which basic information was scattered across several references and sections of the manuscript and supporting information (e.g., number of aerosol sections given in section 3.1.3; the relation of hydrometeor bin sizes to wet or dry particle size in section 3.1.4; an incomplete list of processes given in Table S1). Please define the categories (size, composition, how size is expressed [e.g., wet vs. dry diameter]) and list which moments of the distribution are prognosed (even if only one).

  Regarding the use of references as a stand-in for a model overview in the text/supporting information, a summary of the *current* state of the parameterization is important since I was not able to obtain a preprint of the newest reference (Tonttila et al. 2022, in review, JAMC). Instead, I was forced to go through your code to understand differences between published UCLALES-SALSA configurations and that used in this

paper.

- Two processes are discussed in the last few sections of the paper that I did not see mentioned anywhere earlier on:
(1) Enhancement of collisional growth via turbulence. I could not find any discussion of coupling to turbulence in previous descriptions of SALSA. The code (specifically, mo_salsa_coagulation_kernels.F90) shows that turbulent enhancement of collision-coalescence is tied to a hard-coded eddy dissipation rate rather than responding to local turbulence – and if my assessment of the magnitude is correct, the assumed EDR is rather high and more appropriate for shallow cumulus than stratocumulus (makes sense coming from the Chen et al. 2020 reference). It is also possible that you were discussing some other mechanism for turbulent enhancement of collision-coalescence (i.e., "turbulent fluctuations" at larger scales?), but it was not apparent from the existing discussion. Please clarify.

  (2) Wet scavenging. The last sentence of the paper frames the results of this study as important for the development of wet scavenging schemes in GCMs, but there was no discussion of the impact of wet scavenging on the simulation results. This seems particularly important in the context of Case 2, where sensitivity simulations are performed with aerosol loading artificially reduced by 40% to improve agreement with the in situ drop size distribution and vertical velocity measurements. Did you test the sensitivity of your results to assumed scavenging rates? Could you obtain similar results to the "reduced aerosol loading" experiment by artificially enhancing the rate of scavenging? Given how difficult it is to directly observe scavenging, this seems like an obvious process rate that one might experiment with.

- Presentation of simulation outputs. Let me start by acknowledging: I know your primary focus is on characterizing the aerosol and cloud microphysics output, with a secondary focus on turbulence. So it makes sense that you have many figures comparing size distributions, activation curves, etc., that are difficult to view in a vertically-resolved form. Nevertheless, the lack of profile or curtain-type figures for context on the evolution of the mean thermodynamic and turbulence fields made it difficult to relate to some of the discussion in the text. Specifically, I'd like to see a couple more figures in the spirit of Figures 3 and 9 (time-height curtain plots of cloud LWC and N) for mean thermodynamic fields (temperature and moisture in whatever variables you prefer, perhaps also relative humidity) and turbulence (moments of the w distribution – variance and skewness). You spend some time talking about how the OVL values are validated by obs-model comparisons of variance and skewness, but you never show this. I would imagine that many readers are unfamiliar with the overlapping index metric, so it would be helpful to your argument to make a direct comparison with more "standard" turbulence diagnostics. This would also help with the fact that OVL must be presented on a level-by-level basis; it makes it very difficult to construct a coherent mental image of the vertical development of the turbulence.

MINOR COMMENTS

L37-39: Do your simulations match this heuristic view?

L42-43: This reference to Bougiatoti et al. (2020) is rather over-constrained since it is taken from a single aircraft field campaign over a specific continental region. Are these to be taken as typical values for the whole planet? Shouldn't this be a strong function of boundary layer forcing (surface fluxes, cloud top cooling rate, etc.)? Also, it is not clear that you are discussing respective day vs. night values because of the sentence structure.

L77-78: Other studies have looked at droplet-depletion via turbulence-microphysics interactions; e.g., Remillard et al. (2017) and Witte et al. (2019)

L100: What do you mean by "explicit" calculation? You are not simulating individual particles, so I'm not sure this is appropriate.

L105: Irradiance changes where? The surface? Top of atmosphere? This is an overly vague statement.

L119: here you say the timestepping is leapfrog, but later you say Eulerian-Lagrangian timestepping (L214) – what processes are done with which timestep?

L210: "64 by 64 points" - This is a tiny grid by modern standards. Are you able to spin up realistic turbulence with such a small domain? Also, what is the horizontal grid spacing?

L213: Re: "no significant changes in model outputs" as a function of vertical resolution – Did you compare any measures of DSD spectral width from LES with the observations? There have been several recent studies (e.g., Morrison et al. 2018; Lee et al. 2019, 2021; Witte et al. 2019) that have shown sensitivity of bin microphysics schemes to vertical grid spacing. Given your unique bin arrangement, perhaps it's the case that you arranged your bins similar to the optimal spacing described by Lee et al. (2021), i.e., a combination of linearly-spaced bins for the cloud drop regime and logarithmic bins for the collisionally-dominated regime – but it's not clear from the information you give in the text.

L305: "mass fraction values of…" – The total adds up to more than 100%, so…either go to 3 sig figs or round to reach 100%.

Figure 1 and 2: The markers for Air-Savilahti and Vehmasmäki are indistinguishable without zooming in to 500%. If z=0 is the only level at which Savilahti is displayed, the caption is sufficient to explain data provenance and I suggest you use a single marker for the observations.

Figure 3: This finding re: the increasing trend in N as a function of increased evaporation in cloud is interesting. It seems like such an intuitive consequence of the boundary layer top rising into the dry air above. Very cool.

L391: "significant degree" - Is this in the strict sense of statistical significance, or are you saying it's a large value of OVL?

L420: "moving from fog dynamics to cloud dynamics" – what do you mean? Can you be more specific? I think maybe you're talking about the same phenomenon I was musing on re: my Fig. 3 comment.

L466: what do you mean by "common bin microphysics?" What about the bins is common? The aerosol representation?

L470: "different instruments correlate to each other" -- Correlate, but do not agree. Do you "trust" FM or ICEMET more beyond D>6 µm? They differ by half an order of magnitude in N in Figure 8, although admittedly the modeled DSD has differing agreement as a function of size. Does either instrument have known counting biases in high concentration conditions? Seems like ICEMET should have improved sample volume and optics to detect large particles vs. a forward scattering-based probe.

Figure 8: Whence the secondary peak in the distribution at 20 µm? I think this is where you transition from "cloud" to "drizzle" regimes? Is this just a consequence of the dry vs. wet size difference? A real numerical artifact?

L502: "turbulence was stronger compared to the diurnal case" – isn't this the opposite of what your earlier Bougiatoti et al. (2020) reference said?

L586: compare reduced number concentrations with observed – both are closer to observed case average Nd, worth pointing this out.

L598: "cloud processing is producing larger aerosol particles" – point this out in figures! The info is there in Fig. 13, right?

L622: "but within orders of magnitude below detection limits…" – I'm confused by the word "within" here, does that mean "many" orders of magnitude? Or few? I assume the latter.

L682: Spell out the references to field campaigns, or at least give the appropriate references.

L683: MPACE also used longer-term ARM measurements

SL179-180: "cloud droplets and precipitation droplets" – without knowing these are different categories with different properties, it confusing why you would make this distinction.

TYPOGRAPHICAL COMMENTS

L28: At numerous locations in the manuscript the word "both" is preceded by a comma, i.e. "affecting both, the cloud optical properties and…" – remove these commas.

L33-36: sentence structure does not make sense, in particular the 2nd clause ("droplet formation can be characterized…") – seems like it should be separated from the 1st clause into its own sentence.

L74: "…but it is…disconnected *from* those aerosol…"

L108: "Arctic" instead of "Artic"

L117: "doubly periodic" instead of "doubly periodical"

L125: "cloud hydrometeor*s*"

L141: Capitalize "In situ"

L221: "which were allowed for the second hour before the actual analysis started" – this wording is very confusing. I think you could drop the phrase "before the actual analysis started" and the meaning would be clearer.

L276: "Figure 1 and Figure 2 [show?] the atmospheric…"

L312: "Large particles … promote drizzle formation" (not promotes)

L392: "Halo" – capitalize

L406: Repeat of altitude "(225 m)" – this can be removed, right?

L436: accidental "mum"

L491-492: remove )'s after Fig. 9 references

L559: "local aerosol sources…r*a*ising the probability"

L578-579: suggested rephrasing: "we decided to investigate the extent to which the modelling results…"

L607: I'm not sure "relevant" is the right word here to describe its importance, but I'll leave it up to you whether you want to change it

SL184: "non-activated"

---

## Author Comment (AC1)

**Authors' Response to Reviews of**

**Aerosol-stratocumulus interactions: Towards a better process understanding using closures between observations and large eddy simulations**

Silvia Margarita Calderón, Juha Tonttila, Angela Buchholz, Jorma Joutsensaari, Mika Komppula, Ari Leskinen, Liqing Hao, Dmitri Moisseev, Iida Pullinen, Petri Tiitta, Jian Xu, Annele Virtanen, Harri Kokkola, and Sami Romakkaniemi

*Atmos. Chem. Phys. Discuss.,* `https://doi.org/10.5194/acp-2022-273`
* * *
RC: *Reviewers' Comment*,    AR: Authors' Response,    ☐ Manuscript Text

**Reviewer #1**

**General comment**

**RC:** The authors conducted a closure study to evaluate the UCLALES-SALSA with in situ station measurements during Puijo 2020 campaign. The authors show two cases with different aerosol loadings and meteorology (diurnal v.s. nocturnal) with the boundary layer profiles, cloud macrophysics, and microphysics properties. Sensitivity studies for the second case show that ice processes lead to better agreement between simulated radar velocity and the observed one. The authors highlighted the importance of more observations involving ice particles to reduce the gap in knowledge about the ice-nucleating ability of aerosol particles of both, natural and anthropogenic origin. Overall, I think this is a great manuscript in the closure study of aerosol-cloud interactions. I suggest publishing after the authors address both my key and minor comments below.

**AR:** We deeply appreciate your comments, questions and suggestions. We will proceed to resolve each one of them. In some cases, we have added subsections to address each item.

**1. Key comment**

**RC:** **1.1** I think section 3.3.4 is a very important part of this manuscript, which provides information on the closure of the cloud base velocity. Comparing Figure 11b, Figure 15f, and Figure 16f leads to the conclusion that both reducing the total aerosol loadings and turning on the ice processes leads to better agreements between the model and observations, but the hypothesis cannot be tested due to the lack of observations of ice/mixed phase hydrometeor. Please correct me if my understanding is not what the authors would like to convey. Please confirm Figure 11b is used to compare with Figure 15f and Figure 16f.

**AR:** Yes, Figure 11b, Figure 15f, and Figure 16f must be studied together. Figure 11b compares the distribution of the vertical wind calculated with the model to the distribution of radar velocities observed with the cloud radar (i.e. Hydra-W Millimeter Radar). Figure 11b highlights the displacement to the left of the radar velocity distribution when observations reflect settling velocity of large hydrometeors. Instead, Figure 15f and Figure 16f compare just radar velocities, the observed ones to those estimated from settling velocities of hydrometeors obtained in the simulation scenarios of reduced aerosol loading without and with ice formation, respectively. Therefore, your insight *"Both reducing the total aerosol loadings and turning on the ice processes, lead to*

Table 1: Overlapping indexes between model-based and observation-based distributions of radar velocity in hourly intervals during Case 2 of 31 October 2020. L4: base scenario with no ice formation, L5: basec scenario with ice formation L4-40: simulation scenario with 40% reduction in the aerosol loading used in model initialization without ice formation and related processes, L5-40: simulation scenario with 40% reduction in the aerosol loading including ice formation and related processes.

| Hour | L4 | L5 | L4-40 | L5-40 |
|---|---|---|---|---|
| 1 | 0.800 | 0.709 | 0.778 | 0.726 |
| 2 | 0.748 | 0.898 | 0.791 | 0.907 |
| 3 | 0.847 | 0.757 | 0.943 | 0.818 |
| 4 | 0.704 | 0.919 | 0.816 | 0.908 |
| 5 | 0.677 | 0.909 | 0.836 | 0.925 |
| 6 | 0.618 | * | 0.798 | 0.889 |
| Mean $\pm$ standard deviation | 0.732$\pm$ 0.076 | 0.838$\pm$0.088 | 0.827$\pm$0.055 | 0.862 $\pm$ 0.070 |

*: We did not run the last hour of the simulation

*better agreements between the model and observations, but the hypothesis cannot be tested due to the lack of observations of ice/mixed phase hydrometeor"* conveys our explanation.

RC: **1.2** If I am right, my question *is whether the role of ice phase particles is only significant with low aerosol loading? What will the result look like if not reducing the aerosol loading?*

AR: The role of ice formation is significant in both simulation scenarios, with total and with reduced aerosol loading but more noticeable in the last one. Table 1 summarizes the overlapping index values between model-based and observation-based distributions of the radar velocity for all the simulation scenarios, including the one considering ice related processes without reduction in the aerosol loading used for model initialization. In general, OVL values for the base scenario with ice formation and ice-related processes (L5) indicates an improvement in the description of radar velocities, which suggests improvement in the description of the size and number concentrations of hydrometeors as well as other variables (e.g. aerosol size distribution, activation efficiency curves, droplet size distributions).

RC: **1.3** Also in the text, I don't find any reason to explain the choice of how much reduction in aerosol loading. *How do you come up with the number 40% in line 582? Is there any support for that from either the observation side or previous studies?*

AR: The percentage of reduction in the aerosol loading used for initialization of model calculations (i.e. 40 % reduction) was derived from observations. The beginning of the cloud event of 31, October, 2020 was set at 00:35 (UTC+02:00) when the set in-cloud conditions at the Puijo station were fulfilled (i.e. liquid water content above 0.01 g m$^{-3}$, cloud droplet number concentration higher than 50 cm$^{-3}$ and visibility values below 200 m). Figure 1 shows the variability in the observed aerosol size distributions at the beginning of the cloud event. For the base simulation scenario, we initialized the aerosol properties with the size distribution shown in the left panel, the closest one to 00:35. The relative difference between the total aerosol number concentration in the time period of [00:00–00:32] (Fig. 1, left panel) and that observed in the following time period of [01:00–01:32] (Fig. 1, right panel) is 43%. For the purpose of the model sensitivity analysis, we chose a value of 40% to

[Figure]

Figure 1: Aerosol size distributions measured with the Twin-inlet DMPS system at the beginning of the cloud event of 31,October, 2020. Each panel shows the starting time and the ending time of the observation. Results of the fitting of experimental data to a multi-modal log-normal size distributions are shown for comparison purposes.

reduce the aerosol loading and ran two more simulations, without and with ice-related processes. Although it is very difficult to confirm, the reduction in aerosol loading could be linked to the arrival of cleaner air masses because at 00:40 the wind direction trend changed drastically from NE prevailing winds to NW prevailing winds.

AR: To improve the manuscript, we have added the following sentence to Line 582:

> ... but reduced the total aerosol number concentration used at the model initialization by 40%. This number was taken from the relative difference between measurements of total aerosol number concentrations performed at the beginning of the cloud event (i.e. 00:35) between 00:00 and 00:32 and between 01:00-01:32.

**2. Key comment**

RC: In lines 146-149, the authors said, "THE EFFECT OF LOCAL TOPOGRAPHY ON OBSERVED CLOUD PROPERTIES IS LIMITED TO CERTAIN HIGH WIND CONDITIONS". *What is the definition of a high wind condition? Do the two selected cases associate with high wind or low wind?* The second case has a lower boundary horizontal wind than the first case. *What are the roles of the topography in the clouds examined in these two cases?*.

AR: Romakkaniemi et al. [2017] studied the effects of local topography on cloud droplet formation at Puijo and found that if the wind direction is between $180^o \pm 30^o$, winds above 10 m s$^{-1}$, the air parcels must move through the steepest slope from surface to the Puijo hill interacting with different surface types (e.g. lake, forest). During this trajectory, they can experience a significant degree of convective forcing that induces variability in observed cloud droplet size distributions. In our case studies, the average wind direction was $178.2^o \pm 8.1^o$ and $183.4^o \pm 135.5^o$ for case 1 and case 2, respectively; while the average wind speed was $6.3 \pm 0.6$ m s$^{-1}$ and $3.4 \pm 1.4$ m s$^{-1}$ (quartiles reported in Table S3). The effect of local topography affected more the cloud development in case 1, especially because it started as a fog episode (Fig.03). For this reason we established that our analysis was

focused on the last three hours of the cloud event when the cloud had risen and these effects were no longer significant. For the second case, the wind speed was lower and the wind direction varied between a wider range and we did not notice any particular signal of topographic effects in the aerosol activation into cloud droplets or in the droplet size distribution. We added the following sentence to the manuscript:

> Line 149 ... the effect of local topography on observed cloud properties is limited to certain high wind conditions (i.e. winds above 10 m s$^{-1}$ if the wind direction is $180^o \pm 30^o$ and thus aligned with the steepest slope of the hill)(Romakkaniemi et al., 2017)

**3. Key comment**

**RC:** Figure 1, case 1 is initialized with the boundary layer temperature close to measurements, but the initial temperature above 200m is lower than the observation. *Why not use the observed profile to initialize the model?*

**AR:** The vertical profile of temperature reported in Figure 1 is composed of three datasets representing atmospheric conditions at different altitudes. Observed temperature values above 200 m correspond to measurements from the Vehmasmäki station that it is located 13 km southwest to the Puijo station. There is likely some local variation in the atmospheric profiles and fog top altitude as the surroundings of both Vehmasmäki and Puijo station are heterogeneous. As can be also seen from Figure 3, we were not even targeting to perfectly catch the development of fog/cloud top altitude, but were happy to get the tendency correct with observed surface temperature forcing and estimated temperature profile.

**4. Key comment**

**RC:** In both cases, simulated activation efficiency has a steeper slope than the DMPS observed. The authors stated in lines 452-456, that it is due to the aerosol mixing state. *Can you elaborate on that?.*

**AR:** Both, Fig. SI21 for case 1 and Fig. SI20 for case 2 show steeper slopes for model-based activation efficiency curved than for observation-based. Aerosol composition was simulated as a mixture of sulfate-species and organic carbon-species. In the base simulation scenario for every case, the aerosol population was initialized as an internally mixed set of particles, each one with a constant volume fraction of chemical species throughout the dry particle size range. This in fact, is an oversimplification of the atmospheric real conditions, in which case, the particle hygroscopicity and cloud nucleation ability is both, size and composition dependent. Although in the Case 1 the aerosol is dominated by long range transported aerosol, it still contains a fraction of particles from local sources with varying chemical properties. The differences in the model-based and observation-based activation efficiency curves suggest that in reality a fraction of smaller particles, likely formed during gas-to-particle conversion of sulfate species and therefore are more hygroscopic and susceptible to droplet activation than those represented by the model. Likewise, experimental curves suggest that some large particles are more likely composed of mixtures of primary aerosols with the lower hygroscopicity. This type of size-dependent hygroscopicity of aerosol particles is translated into weaker slopes of the experimental activation efficiency curves. The fraction of activated particles increases with increasing size at a lower rate for larger particles that are less hygroscopic [e.g. Anttila, 2010]. Thus, by assuming in our modelling framework that the aerosol population is internally-mixed, we are using the behaviour of the particle with the average hygroscopicity (or the most and less hygroscopic particles if the aerosol is assumed to be externally mixed) as

surrogate of all aerosol particles. This indeed, guides the modeled activation efficiency curves towards the observed $D_{50}$ (see Fig. SI21 for case 1 and Fig. SI20 for case 2) but induces deviations from the observed behaviour for dry particle sizes below and above the $D_{50}$. A more detailed modelling of the size-dependence hygroscopicity is computationally too expensive in a LES model. In Fig. SI21 we show how modeled activation efficiency curves calculated for an externally mixed aerosol population consisting of two populations were in better agreement to the observed ones, and also how differences between modeled curves for internal and external mixing states of the aerosol population decrease with stronger and more frequent updrafts. However, as it was mentioned in the manuscript, the treatment of the particle mixing state did not have significant effects in overall droplet number concentrations or size distributions.

AR: The following text was added to the manuscript:

> Since Case 1 occurred during the biomass burning plume period, it is likely to have an externally mixed aerosol population composed of two types of particles, particles locally emitted or formed in situ, and particles from aged biomass burning emissions transported long range. Unfortunately, measurements do not provide information on the aerosol mixing state. Despite that, to assess the potential effect of the aerosol chemical diversity in our simulations, we compared the simulation results obtained for an internally mixed aerosol population with those for an externally mixed aerosol population with the same aerosol number size distribution. As expected, the slopes in activation efficiency curves of the externally mixed aerosol population were less steep than those for the internally mixed aerosols and match better the observed slopes. Nevertheless, there were no significant changes in $D_{50}$ values nor in droplet number concentrations and size distributions. The differences in the model-based and observation-based activation efficiency curves suggest that in reality there was a fraction of smaller particles, likely formed during gas-to-particle conversion of sulfate species that was more hygroscopic and susceptible to droplet activation than those represented by the model.. Detailed information is included in Section 9 of the supporting information.

RC: **4.2** My understanding is external mixing leads to a higher insoluble fraction of aerosols so that the activation of particles is suppressed. But, *why large particles are suppressed more than smaller particles? Is external mixing on large particles expected more frequent than on smaller particles?* I think a more detailed explanation is needed and literature is needed to explain this, although there is no observation to provide the mixing state of aerosols.

AR: Without size dependent growth factor observations with HTDMA, like we presented earlier in the Vaisanen et al (2016), it is difficult to answer to your question "Is external mixing on large particles expected more frequent than on smaller particles?". From the modelling point of view, external mixing suppress activation because it leads to a higher insoluble fraction (lower soluble fraction) of aerosols without notable change in the overall condensation rate and thus supersaturation reached. The soluble fraction is proportional to the third power of the growth factor and therefore it is very sensitive to changes in the aerosol hygroscopicity [e.g. Malm and Kreidenweis, 1997, Swietlicki et al., 2008]. In a very simplistic approach based on the Köhler theory, the critical supersaturation is modified via the Raoult term because the soluble fraction decreases as well as the effective number of solute moles in the droplet solution in direct proportion to the particle size. However, the magnitude of these effects on small and large particles depend also on the degree of mixing and the numerical treatment given to the particle mixing state [e.g. Anttila, 2010]

**5. Key comment**

**RC:** In Figure 8, the Rotating holographic imager shows a higher concentration on the large particle, and the fog monitor shows a smaller concentration between about 6-25 microns. These differences are not changed at both 3 hrs and 5 hrs. As a closure study, *what is the reason for the differences= Model uncertainties or observation uncertainties?*

**AR:** The rotating holographic imager (ICEMET) and the fog monitor have different observational ranges. Their performance during the Puijo 2020 was analyzed in an intercomparison study by Tiitta et al. [2022]. Differences in hydrometeor number concentrations correspond mainly to sampling problems for the fog monitor. Unlike the ICEMET (rotating holographic imager) that has a vane and rotates accordingly to the prevailing winds, the fog monitor inlet is stationary, and during the sampling campaign, it was set to face east ($90^o$). The intercomparison study found that the fog monitor experienced droplet losses due to anisoaxial sampling when prevailing winds during sampling were outside of the interval $90^o \pm 30^o$. These losses had a significant effect on the liquid water content measured by the fog monitor in particular time intervals.For the cloud event of 24, September, 2020; the average wind direction was $178.2^o \pm 8.1^o$ and the fog monitor experienced a systematic loss of large droplets (i.e. droplet diameter above 6 $\mu$m) because of anisoaxial sampling.

**Minor comments**

**RC:** Figure 4 and Figure 10, change the color of the dark blue bar, otherwise, it is hard to tell the differences between UCLALES-SALSA updrafts with Halo Doppler lidar. Figure 5 and Figure 11 are good examples.

**AR:** Thanks for the suggestion. Color schemes in all figures were changed and hopefully improved. Figure 2 is an example of the suggested change.

**RC:** Line 436, change mum to $\mu$m

**AR:** Thank you for noticing the typo.

> Line 436: ... we calculated $SS_{eff}$ based on $D_{50}$ values obtained with a cut off size of 2  $\mu$m

**RC:** Line 472, correct the format of the citation.

**AR:** Thanks.

> Line 474: ... the performance of both instruments during the Puijo 2020 sampling campaign are provided in  Tiitta et al. (2022)

**References**

S. Romakkaniemi, Z. Maalick, A. Hellsten, A. Ruuskanen, O. Väisänen, I. Ahmad, J. Tonttila, S. Mikkonen, M. Komppula, and T. Kühn. Aerosol–landscape–cloud interaction: signatures of topography effect on cloud droplet formation. *Atmospheric Chemistry and Physics*, 17(12):7955–7964, 2017. . URL https://acp.copernicus.org/articles/17/7955/2017/.

[Figure]

Figure 2: New color scheme: Comparison of model-based distributions of vertical wind at in-cloud conditions during Case 1 24 September 2020 to those retrieved from cloud radar observations. Each panel shows the overlapping index value (OVL) as an indicator of agreement between distributions.

Tatu Anttila. Sensitivity of cloud droplet formation to the numerical treatment of the particle mixing state. *Journal of Geophysical Research*, 115(D21):D21205, nov 2010. ISSN 0148-0227. . URL http://doi.wiley.com/10.1029/2010JD013995.

William C Malm and Sonia M Kreidenweis. The effects of models of aerosol hygroscopicity on the apportionment of extinction. *Atmospheric Environment*, 31(13):1965–1976, 1997. ISSN 1352-2310. . URL http://www.sciencedirect.com/science/article/pii/S135223109600355X.

E Swietlicki, H C. Hansson, K Hämeri, B Svenningsson, A Massling, G Mcfiggans, P H Mcmurry, T Petäjä, P Tunved, M Gysel, D Topping, E Weingartner, U Baltensperger, J Rissler, A Wiedensohler, and M Kulmala. Hygroscopic properties of submicrometer atmospheric aerosol particles measured with H-TDMA instruments in various environments—a review. *Tellus B: Chemical and Physical Meteorology*, 60(3): 432–469, 2008. . URL https://doi.org/10.1111/j.1600-0889.2008.00350.x.

P Tiitta, A Leskinen, V Kaikkonen, E Molkoselkä, A Mäkynen, J Joutsensaari, S Calderon, S Romakkaniemi, and M Komppula. Intercomparison of holographic imaging and single-particle forward light scattering in-situ measurements of liquid clouds in changing atmospheric conditions. *Atmospheric Measurement Techniques Discussions*, 2022:1–20, 2022. . URL https://amt.copernicus.org/preprints/amt-2021-423/.

**Authors' Response to Reviews of**

**Aerosol-stratocumulus interactions: Towards a better process understanding using closures between observations and large eddy simulations**

Silvia Margarita Calderón, Juha Tonttila, Angela Buchholz, Jorma Joutsensaari, Mika Komppula, Ari Leskinen, Liqing Hao, Dmitri Moisseev, Iida Pullinen, Petri Tiitta, Jian Xu, Annele Virtanen, Harri Kokkola, and Sami Romakkaniemi

*Atmos. Chem. Phys. Discuss.,* `https://doi.org/10.5194/acp-2022-273`
* * *
RC: *Reviewers' Comment*,     AR: Authors' Response,     ☐ Manuscript Text

**Reviewer #2**

**Summary**

**RC:** This study describes observations and simulations of aerosol-cloud interactions via two case studies of low stratiform cloudiness at a continental site in eastern Finland. The cases are characterized by a unique suite of measurements similar to a US DOE Atmospheric Radiation Measurement program site including in situ measurements of basic meteorological variables (pressure, temperature, horizontal winds, water vapor amount) as well as aerosol and cloud properties; and remote sensing of cloud and boundary layer dynamical properties (i.e., vertical velocity) from a W-band Doppler radar-radiometer system, ceilometer and Doppler lidar. The simulations are designed to simulate aerosol-cloud interactions in a highly detailed manner using large eddy simulations coupled to a size-resolved Eulerian aerosol/cloud microphysics framework. The overall goal of the study is to demonstrate the significant degree of microphysical closure that can be obtained with the modeling framework. A vast (and impressive) amount of observational information was synthesized to provide a comprehensive reference for the simulations and while the purpose of the study was to demonstrate the performance of the model, there was relatively little information given on the model construction and configuration. What model information was given was rather spread out over the manuscript, which made it difficult to understand exactly how the simulations were run. In addition, several major points were made in the conclusions/discussion regarding microphysical mechanisms that were barely mentioned in the results and relevant figures – this felt rather incongruous. As a result, I recommend the manuscript be returned to the authors for major revisions.

AR: We deeply appreciate the detailed-oriented review of this study. We had rephrased some sentences in the manuscript to reflect better our research goal that is "to simulate aerosol-cloud interactions in a highly detailed manner using large eddy simulations coupled to a size-resolved Eulerian aerosol/cloud microphysics framework".

AR: Many of the important issues highlighted by the reviewer are addressed in detail in the next sections.

**1. Major comment**

**RC:** The study builds upon a large body of previous and contemporaneous work to define and characterize the case studies and describe the modeling framework. In general, I found that the authors relied so heavily on references to other works that I was not able to understand much about their approach without close reading of these other manuscripts. I feel strongly that the paper should include further relevant details such that it can stand on its own. This is particularly the case with respect to the microphysics scheme, for which basic information was scattered across several references and sections of the manuscript and supporting information (e.g., number of aerosol sections given in section 3.1.3; the relation of hydrometeor bin sizes to wet or dry particle size in section 3.1.4; an incomplete list of processes given in Table S1). Please define the categories (size, composition, how size is expressed [e.g., wet vs. dry diameter]) and list which moments of the distribution are prognosed (even if only one).

**AR:** We understand your point. We focused our attention on the description of the pairing process of experimental and modeled data because UCLALES-SALSA has been available for the community of atmospheric modelling since 2017 [Tonttila et al., 2017] and used in several studies of atmospheric processes [e.g. Stevens et al., 2018, Ahola et al., 2020, Tonttila et al., 2021, Slater et al., 2021, 2022, Raatikainen et al., 2022]. However, we have added a detailed description of the bin scheme for aerosol particles and hydrometeors used by UCLALES-SALSA to the supporting information. The list of variables with their correspondent description and units is available as metadata [Calderón et al., 2022]. Section 2.4 "Model setup" was modified as follows:

> Initial conditions for size-segregated aerosol number concentrations were fed into the model as multi-modal lognormal functions $n_N(D_p)$ with parameters fitted to measurements taken with the Twin-inlet DMPS system from the total inlet at the beginning of each cloud event, 24 September 2020 07:54 (UTC+2) and 31 October 2020 00:35 (UTC+2). Parameters for size distributions are reported in Table 1. The employed bin scheme includes 18 size bins for both two mixing states for aerosol particles (i.e. regime A and regime B), 15 size bins for cloud droplets generated from both aerosol regime, 20 size bins for drizzle/rain droplets and 20 size bins for ice particles. Size bins for aerosols (non activated droplets) and cloud droplets (activated droplets) are referred to the dry state. Wet diameters for each categories are stored separated variables. Size bins for precipitation droplets and ice particles are expressed in wet diameter. Details on the bin grid are included in Section 1 of the supporting information. Aerosol particles were assumed to be internally mixed.

**2. Major comment**

**RC:** Regarding the use of references as a stand-in for a model overview in the text/supporting information, a summary of the current state of the parameterization is important since I was not able to obtain a preprint of the newest reference (Tonttila et al. 2022, in review, JAMC). Instead, I was forced to go through your code to understand differences between published UCLALES-SALSA configurations and that used in this paper.

**AR:** We apologized for this unexpected inconvenient. The paper of Tonttila et al., (2022) is in the final stage before publication and will be available in days. As mentioned before, we have improved the description of the modelling framework in the supporting information.

**3. Major comment**

**RC:** Two processes are discussed in the last few sections of the paper that I did not see mentioned anywhere earlier on:" (1) Enhancement of collisional growth via turbulence. I could not find any discussion of coupling to turbulence in previous descriptions of SALSA. The code (specifically, mo_salsa_coagulation_kernels.F90) shows that turbulent enhancement of collision-coalescence is tied to a hard-coded eddy dissipation rate rather than responding to local turbulence – and if my assessment of the magnitude is correct, the assumed EDR is rather high and more appropriate for shallow cumulus than stratocumulus (makes sense coming from the Chen et al. 2020 reference).

**AR:** First of all, we must express our gratefulness for your insightful observation. You are correct. We did indeed run our simulations with a constant eddy dissipation rate of $1 \times 10^{-2}$ m$^2$ s$^{-3}$, a value that is somewhat higher than those retrieved from radar observations of stratocumulus clouds [e.g. Fang et al., 2014] and closer to mean values observed in shallow cumulus (Cu humilis) [e.g. Siebert et al., 2006, Chen et al., 2020]. We are aware of this model deficiency and will tackle it in future work. However, in this study, we took advantage of the agreement between experimental and modeled droplet size distributions to evaluate the assumed value for the eddy dissipation rate.

**RC:** Continue from "Two processes are discussed in the last few sections of the paper that I did not see mentioned anywhere earlier on:" (1) Enhancement of collisional growth via turbulence... It is also possible that you were discussing some other mechanism for turbulent enhancement of collision-coalescence (i.e.,"turbulent fluctuations" at larger scales?), but it was not apparent from the existing discussion. Please clarify.

**AR:** You are correct. We refer to the enhancement of collision-coalescence by turbulent fluctuations at larger scales. We have rephrased some sentences to make clear this important point.

**AR:** In the introduction:

> The strength of these large scale turbulent circulations is further enhanced by the gas-liquid energy exchange during condensation processes in updrafts and evaporation and cooling in downdrafts (Wood, 2012).

**AR:** In section 3.1.3 Size-dependent activation efficiency:

> To assess the effect of large scale turbulent circulation , we studied separately the activation efficiency in grid points with updraft winds or downdraft winds.

**AR:** In section 3.3.4 Model sensitivity analysis to inputs of aerosol number concentrations in simulations of Case 2 :

> However, with larger droplet sizes, collisional growth is also more efficient and become enhanced by large scale turbulent circulation (Yang et al., 2018; Chen et al., 2018b) and positive buoyant fluxes locally induced by droplet sedimentation (Mellado, 2017).

**RC:** Continue from "Two processes are discussed in the last few sections of the paper that I did not see mentioned anywhere earlier on:" (2) Wet scavenging. The last sentence of the paper frames the results of this study as important for the development of wet scavenging schemes in GCMs, but there was no discussion of the impact of wet scavenging on the simulation results. This seems particularly important in the context of Case 2, where sensitivity simulations are performed with aerosol loading artificially reduced by 40% to improve agreement with the in situ drop size distribution and vertical velocity measurements. Did you test the sensitivity of

your results to assumed scavenging rates? Could you obtain similar results to the "reduced aerosol loading" experiment by artificially enhancing the rate of scavenging? Given how difficult it is to directly observe scavenging, this seems like an obvious process rate that one might experiment with.

AR: The scavenging rate can be derived via coagulation rates. However, we did not calculate this variable during our simulation as it was not an specific research goal for this study. We removed the sentence from the manuscript to avoid confusion. However with the current settings and model outputs included in the metadata [Calderón et al., 2022] for this study, we could calculate variations in total aerosol mass related to scavenging using the bulk mass of aerosol constituents present in aerosol particles, cloud droplets and precipitation droplets (i.e. model outputs listed as aOCa,cOCa,pOCa, aSO4, cSO4, pSO4).

> As the effect of cloud processing on aerosol properties is difficult to constrain using observations, the findings support the further employment of models like UCLALES-SALSA for the development of wet scavenging schemes accounting for different chemical compounds in global models.

**4. Major comment**

RC: Presentation of simulation outputs. Let me start by acknowledging: I know your primary focus is on characterizing the aerosol and cloud microphysics output, with a secondary focus on turbulence. So it makes sense that you have many figures comparing size distributions, activation curves, etc., that are difficult to view in a vertically-resolved form. Nevertheless, the lack of profile or curtain-type figures for context on the evolution of the mean thermodynamic and turbulence fields made it difficult to relate to some of the discussion in the text. Specifically, I'd like to see a couple more figures in the spirit of Figures 3 and 9 (time-height curtain plots of cloud LWC and N) for mean thermodynamic fields (temperature and moisture in whatever variables you prefer, perhaps also relative humidity) and turbulence (moments of the w distribution – variance and skewness).

AR: Curtain-type plots for the air temperature and specific humidity are presented below in Figure 1 and Figure 2.

AR: Curtain-type plots for the variance of the emulated radar velocity are included in Figure 3 and compared to values retrieved from cloud radar observations. Using a more conventional approach, Figure 4 shows the degree of agreement between model-based and radar-based event-average values for the variance of the vertical wind distribution. A similar comparison for the skewness is not appropriate since this statistical parameter is very sensitive to extreme values. Extreme values are here present in observations likely as a result of local surface topography. Skewness can also be affected by the simultaneous presence of liquid droplets and ice particles especially in Case 2 (Figure 5) as its sign depends on the phase that dominates the signal and the phase partitioning can change rapidly with time.

RC: You spend some time talking about how the OVL values are validated by obs-model comparisons of variance and skewness, but you never show this. I would imagine that many readers are unfamiliar with the overlapping index metric, so it would be helpful to your argument to make a direct comparison with more "standard" turbulence diagnostics. This would also help with the fact that OVL must be presented on a level-by-level basis; it makes it very difficult to construct a coherent mental image of the vertical development of the turbulence.

AR: We are aware of the fact the overlapping index (OVL) value is not commonly used in atmospheric studies. However, this statistical parameter was extremely useful to summarize the agreement between the distributions and has the advantage of being less susceptible to extreme values. For example, Figure 6 and Figure7 show

[Figure]

Figure 1: Vertical profile of air temperature and specific humidity calculated with UCLALES-SALSA for the cloud event of 24, September, 2020

[Figure]

Figure 2: Vertical profile of air temperature and specific humidity calculated with UCLALES-SALSA for the cloud event of 31, October, 2020

[Figure]

Figure 3: Vertical profile of the variance in the radar velocity calculated with UCLALES-SALSA for the studied cases. Variance for the observed radar velocity in time intervals of 20 min are shown for comparison purposes.

[Figure]

Figure 4: Variance and skewness of the event-average vertical wind distribution calculated with UCLALES-SALSA and retrieved from cloud radar observations for Case 1

[Figure]

Figure 5: Variance and skewness of the 4h-average vertical wind distribution calculated with UCLALES-SALSA and retrieved from cloud radar observations for Case 2. Simulation scenario with reduction in aerosol loading without ice related processes

how a high overlapping index (e.g. OVL > 0.9) indicates good agreement between the variance and skewness of the model-based and observation-based distributions.

**RC:** OVL must be presented on a level-by-level basis; it makes it very difficult to construct a coherent mental image of the vertical development of the turbulence.

**AR:** Figure 8 and Figure 9 summarize the spatial and time variability of the overlapping index between model-based and observation-based distributions of radar velocity for the selected cloud events.

**AR:** Despite adding these figures to this discussion, these plots were not included in the manuscript nor the supporting information since we consider that they do not provide significant new information to the reader.

**5. Minor comments**

**5.1.**

**RC:** L37-39: Do your simulations match this heuristic view?

**AR:** The dynamics of cloud formation is so complex that it would be very bold to say that our study was planned to show a heuristic view of this phenomenon. Cloud events were selected on the basis of simpler criteria." L206 : Selected events reflect contrasting scenarios of cloud formation in terms of the aerosol loading and turbulence driving mechanism (e.g. cloud radiative cooling vs. surface heating, updraft-limited regime vs. aerosol-limited regime). Case 1 occurs under surface-driven turbulence with high aerosol loading, while case 2 occurs under

[Figure]

Figure 6: Characteristic parameters of the distribution of the vertical wind measured under different metrics

[Figure]

Figure 7: Characteristic parameters of the distribution of the vertical wind measured under different metrics

[Figure]

Figure 8: Overlapping index between model-based distributions of the vertical wind and observations from the cloud radar during the cloud event of 24, September, 2020

[Figure]

Figure 9: Overlapping index between model-based distributions of the vertical wind and observations from the cloud radar during the cloud event of 31, October, 2020

cloud-top driven turbulence and low aerosol loading.

**5.2.**

**RC:** L42-43: This reference to Bougiatoti et al. (2020) is rather over-constrained since it is taken from a single aircraft field campaign over a specific continental region. Are these to be taken as typical values for the whole planet? Shouldn't this be a strong function of boundary layer forcing (surface fluxes, cloud top cooling rate, etc.)? Also, it is not clear that you are discussing respective day vs. night values because of the sentence structure.

**AR:** Yes, you are right. The reference was removed in the manuscript.

>

**5.3.**

**RC:** L77-78: Other studies have looked at droplet-depletion via turbulence-microphysics interactions; e.g., Remillard et al. (2017) and Witte et al. (2019)

**AR:** Thank you very much. Studies by Rémillard et al. [2017a,b] were in-line with our comments and were included in the manuscript.

**5.4.**

**RC:** L100: What do you mean by "explicit" calculation? You are not simulating individual particles, so I'm not sure this is appropriate.

**AR:** We meant that the model includes an specific treatment for the size and composition of aerosol particles, intimately linked to the microphysical processes. For example, the aerosol growth through condensation is used to assess the droplet activation, instead of using a parameterization or predetermined CCN concentrations. To avoid confusion we have modified the sentence as follows:

> UCLALES-SALSA is a large eddy simulation model  with a size-resolved description of particle compositions and microphysical processes in aerosol and clouds. This detailed representation allows for example to use aerosol growth though condensation to assess the droplet activation, instead of recurring to parametrizations or having pre-determined CCN concentrations.

**5.5.**

**RC:** L105: Irradiance changes where? The surface? Top of atmosphere? This is an overly vague statement.

**AR:** You are right. It is very difficult to describe in a short way the abilities of LES models given their complexity and level of detail in the process description. We meant that UCLALES-SALSA is a versatile modelling framework that allows for studying changes in radiation fluxes caused by aerosol-cloud interactions with small-scale meteorology at any section of the model domain. We removed the sentence in the manuscript.

>

**5.6.**

**RC:** L119: here you say the timestepping is leapfrog, but later you say Eulerian-Lagrangian timestepping (L214) – what processes are done with which timestep?

**AR:** The prognostic momentum equations use leapfrog timestepping scheme and prognostic scalars use simple eulerian forward timestepping

**5.7.**

**RC:** L210: "64 by 64 points" - This is a tiny grid by modern standards. Are you able to spin up realistic turbulence with such a small domain? Also, what is the horizontal grid spacing?

**AR:** We must apologize since the value used as horizontal resolution in our simulations was unintentionally missing. The horizontal resolution is 30 m (i.e. $\Delta X$ and $\Delta Y$ are 30 m), while the vertical resolution is 10 m. This gives a model domain of 1920 m $\times$ 1920 m $\times$ 1200 m, which is indeed, small compared to the typical size handled by LES based on bulk microphysics schemes. However, as also the boundary layer height in studied cases is very low, we are confident that our simulations captured the dynamics of cloud formation as suggested by the agreement between observed and modeled macrophysical cloud properties (e.g. cloud boundaries retrieved from radar observations and modeled liquid water content profiles). The following modification was performed to the section entitled Model setup:

[Figure]

Figure 10: Spectral dispersion parameter calculated with UCLALES-SALSA and retrieved from observations of the fog monitor and the rotating holographic imaging system

> The model domain comprised a horizontal grid of 64 by 64 equidistant points separated by 30 m with a vertical grid extended up to an altitude equivalent to three times the cloud top height retrieved from radar profiles (i.e. 1200 m).

**AR:** As we do not expect the studied clouds to have mesoscale organization affecting the observed cloud properties, we focused on the aerosol-cloud-interactions and therefore, devoted the computational resources and time to have a very fine sectional representation that includes 18 size bins in two mixing states for aerosol particles (i.e. regime A and regime B), 15 size bins for cloud droplets generated from each aerosol regime, 20 size bins for drizzle/rain droplets and 20 size bins for ice particles totalling over 200 aerosol tracers even for the most simple case studied..

**5.8.**

**RC:** L213: Re: "no significant changes in model outputs" as a function of vertical resolution – Did you compare any measures of DSD spectral width from LES with the observations? There have been several recent studies (e.g., Morrison et al. 2018; Lee et al. 2019, 2021; Witte et al. 2019) that have shown sensitivity of bin microphysics schemes to vertical grid spacing.

**AR:** Tonttila et al. [2021] tested the effect of changing the vertical resolution on the simulated precipitation rate for a marine stratocumulus cloud case (Eastern Pacific Emitted Aerosol Cloud Experiment E-PEACE, 2011) [Jung et al., 2015] using UCLALES-SALSA. There were minor differences in the mean precipitation flux close to cloud base when the vertical resolution was reduced from 10 m to 5 m. We found the same results at the preliminary stage in our study and decided to keep 10 m as the vertical resolution to deviate all computational resources to the representation of aerosol and hydrometeors microphysics. We compared model outputs to observations for count mean diameter, surface mean diameter, effective diameter and spectral dispersion parameter and found a reasonable agreement as shown in Figure 10. However, we did not include them in the manuscript or the supporting information to limit the number of figures in total.

**RC:** Given your unique bin arrangement, perhaps it's the case that you arranged your bins similar to the optimal spacing described by Lee et al. (2021), i.e., a combination of linearly-spaced bins for the cloud drop regime and logarithmic bins for the collisionally-dominated regime – but it's not clear from the information you give

[Figure]

Figure 11: Radius bin grid spacing as a function of radius in UCLALES-SALSA compared to the optimal spacing described by Lee et al. [2021] for different parameters.

in the text.

**AR:** Figure 11 shows the optimal bin grid to represent droplet microphysics of Lee et al. [2021] and also the bin grids used by UCLALES-SALSA. While for the precipitation droplets the bin grid is referred to the wet size, bin grids for aerosol particles (i.e. non activated particles) and cloud droplets (i.e. activated particles with droplet diameter below 20 $\mu$m) are referred to the dry size and therefore are not directly comparable to the Lee's optimal bin grid. The wet sizes for each aerosol bin and cloud droplet bin are stored in individual multi-dimensional arrays (i.e.z,x,y,bin,time). Rates of all microphysical processes involving aerosol-hydrometeor or hydrometeor-hydrometeor interactions are calculated using these wet sizes. When the size of the droplet formed in a collision between two cloud droplets size exceeds a limiting size of 20 $\mu$m, the droplet is moved to the correspondent bin in the grid for precipitation (drizzle/rain) droplets to allow for realistic representation of droplet growth into precipitating sizes.

**5.9.**

**RC:** L305: "mass fraction values of..." – The total adds up to more than 100%, so...either go to 3 sig figs or round to reach 100%.

AR: Thank you. There was a typo in the sentence.

> Aerosol composition varied more than that in Case 1 with average mass fraction value of 47.3 ± 23.1 % w/w for sulfate species. .

**5.10.**

RC: Figure 1 and 2: The markers for Air-Savilahti and Vehmasmäki are indistinguishable without zooming in to 500%. If z=0 is the only level at which Savilahti is displayed, the caption is sufficient to explain data provenance and I suggest you use a single marker for the observations.

AR: Thank you. This change improved the figures.

**5.11.**

RC: Figure 3: This finding re: the increasing trend in N as a function of increased evaporation in cloud is interesting. It seems like such an intuitive consequence of the boundary layer top rising into the dry air above. Very cool.

AR: Thanks.

**5.12.**

RC: L391: "significant degree" - Is this in the strict sense of statistical significance, or are you saying it's a large value of OVL?

AR: The "significant degree" term was removed from the sentence to avoid any confusion with the statistical significance. We meant that a larger OVL value indicates more overlapping between distributions.

**5.13.**

RC: L420: "moving from fog dynamics to cloud dynamics" – what do you mean? Can you be more specific? I think maybe you're talking about the same phenomenon I was musing on re: my Fig. 3 comment.

AR: As the fog, by definition, is in contact with the surface, the activation of aerosol particles to cloud droplets is less efficient than at the cloud base where updrafts are stronger. Also, in fog there are pre-existing fog droplets that decrease the water supersaturation and thus limit the formation of new fog droplets. This has been studied for example in Tonttila et al. [2017] and Boutle et al. [2018] in more detail. We modified the manuscript in this way:

> This change is likely due to an increase in the cloud base altitude, and moving from fog dynamics to cloud dynamics with higher maximum supersaturations and thus more efficient droplet formation.

**5.14.**

RC: L466: what do you mean by "common bin microphysics?" What about the bins is common? The aerosol representation?

AR: By common bin microphysics, we meant that the sectional representation of cloud droplet size distribution is set to have the same bin limiting values within the common size range with respect to the aerosol size distribution [Tonttila et al., 2017]. When the wet diameter of liquid droplets formed in the coagulation between

[Figure]

Figure 12: Bin scheme of UCLALES-SALSA for aerosol particles and hydrometeors

cloud droplets exceeds a limiting value of 20 $\mu$m, the resulting droplet is moved to the proper size bin in the sectional scheme for precipitation droplets. Ice particles are always located in the ice particle bins where minimum size corresponds the spherical equivalent diameter of 2 $\mu$m. Size distributions are built using volume ratio discretization [Jacobson, 2005]. Figure 12 describes the relationships between the schemes used for aerosol and hydrometeors. Whilst aerosol and cloud droplets have parallel size bins, precipitation droplets and ice particles have their own scheme.

AR: The following text was added to the manuscript in Section 2.1 UCLALES-SALSA modelling framework:

> Aerosol particles are separated into non-activated and activated particles depending on water supersaturation and wet size of particles, and then redistributed among size bins between interstitial aerosol and cloud droplets. The sectional representation of aerosol particles and cloud droplets is based on dry size and shares the same bin limiting values within a common size range. Wet sizes of aerosol particles and all hydrometeors are stored separately in reference to their common microphysics based on dry size. When the wet diameter of a liquid droplet exceeds a limiting value of 20 $\mu$m, the droplet is moved to the proper size bin in the sectional scheme for precipitation droplets. Ice particles are always located in the ice particle bins where minimum size corresponds the spherical equivalent diameter of 2 $\mu$m. Sectional schemes for precipitation droplets and ice particles are built using volume ratio discretization (Jacobson, 2005).. More information about aerosol size and composition and bin schemes can be found in the original SALSA description by by Kokkola et al. (2008, 2018); Tonttila et al. (2017); Ahola et al. (2020); Tonttila et al. (2021). Microphysics of liquid droplets was explained by Tonttila et al. (2017, 2021) while ice microphysics was described by Ahola et al. (2020); Tonttila et al. (2022). Section 1 of the supporting information includes details of modelling frameworks used for each one of the microphysical processes.

AR: The following text was added to Section 3.1.4:

> Since UCLALES-SALSA uses  bin microphysics based on dry particle size for aerosol particles and cloud dropletsbefore performing any averaging operation, it was necessary to group hydrometeors into size-resolving microphysics based on wet size using the correspondent model outputs for droplet diameter. Consecutive size bins for wet size have a volume ratio of 2-3 with values ranging from 0.5 $\mu$m to 2 mm.

**5.15.**

**RC:** L470: "different instruments correlate to each other" – Correlate, but do not agree. Do you "trust" FM or ICEMET more beyond D>6 μm? They differ by half an order of magnitude in N in Figure 8, although admittedly the modeled DSD has differing agreement as a function of size. Does either instrument have known counting biases in high concentration conditions? Seems like ICEMET should have improved sample volume and optics to detect large particles vs. a forward scattering-based probe.

**AR:** The rotating holographic imager (ICEMET) and the fog monitor have different observational size ranges without counting biases in high concentrations conditions. Their performance during the Puijo 2020 was analyzed in an intercomparison study by Tiitta et al. [2022]. Differences in hydrometeor number concentrations correspond mainly to sampling problems for the fog monitor. Unlike the ICEMET (rotating holographic imager) that has a vane and rotates accordingly to the prevailing winds, the fog monitor inlet is stationary, and during the sampling campaign, it was set to face east ($90^o$). The intercomparison study found that the fog monitor experienced droplet losses due to anisoaxial sampling when prevailing winds during sampling were outside of the interval $90^o \pm 30^o$. These losses had a significant effect on the liquid water content measured by the fog monitor in particular time intervals.For the cloud event of 24, September, 2020; the average wind direction was 178.2$^o$±8.1$^o$ and the fog monitor experienced a systematic loss of large droplets (i.e. droplet diameter above 6 um) because of anisoaxial sampling.

**AR:** The intercomparison study of Tiitta et al. [2022] showed that measurements of liquid water content (LWC) of ICEMET were comparable to the estimated theoretical maximum LWC, and more reliably than those from the FM-120 due to the relevant influence of the fog monitor inlet (i.e no swivel-head mount.

**5.16.**

**RC:** Figure 8: Whence the secondary peak in the distribution at 20 μm? I think this is where you transition from "cloud" to "drizzle" regimes? Is this just a consequence of the dry vs. wet size difference? A real numerical artifact?

**AR:** Yes, the peak at 20 $\mu$m in the droplet size distribution shown in Fig. 8 corresponds to the transition of liquid droplets from the cloud droplet scheme to the precipitation (i.e. drizzle and rain droplets). It is a numerical artefact related to the limiting size of evaporating precipitating droplet below which where they are moved from droplet to aerosol particle bins. We explored different options to smooth the distribution at this size range, but that artefact is stubbornly visible in these cases where cloud base height is low and thus relative humidity even at the surface is high. Thus it is possible that droplets in the smallest precipitation bin are circulated back to cloud and in such case there is accumulation of droplets on that bin. That does not affect the cloud thermodynamics as all microphysical processes are calculated similarly for both cloud and precipitation bins.

**5.17.**

**RC:** L502: "turbulence was stronger compared to the diurnal case" – isn't this the opposite of what your earlier Bougiatoti et al. (2020) reference said?

**AR:** You are right. The statement referring to Bougiatoti et al. (2020) was confusing and was removed from the manuscript. However, in Figure 3 we notice that the variance of the vertical wind velocity moves in a small range between 0.25 and 0.3 during the nocturnal cloud or case 2, while it increases monotonically with time up to a maximum of 0.3 approximately at the end of the diurnal cloud or case 1.

**5.18.**

**RC:** L586: compare reduced number concentrations with observed – both are closer to observed case average Nd, worth pointing this out.

**AR:** Thank you. This was indeed necessary. The percentage of reduction in the aerosol loading used for initialization of model calculations (i.e. 40 % reduction) was derived from observations. The beginning of the cloud event of 31, October, 2020 was set at 00:35 (UTC+02:00) when in-cloud conditions at the Puijo station were fulfilled (i.e. liquid water content above 0.01 g m$_{-3}$, cloud droplet number concentration higher than 50 cm$^{-3}$ and visibility values below 200 m). The selected value for aerosol reduction comes from the relative difference between the total aerosol number concentration from consecutive measurements performed in the intervals of [00:00-00:32] and [01:00–01:32]. At 00:40 the wind direction trend changed from prevailing wind from NE to prevailing winds from NW. There is more information about this in the response to the key comment 1.1 from the reviewer 1.

**5.19.**

**RC:** L598: "cloud processing is producing larger aerosol particles" – point this out in figures! The info is there in Fig. 13, right?

**AR:** You are right, Fig. 13 implicitly includes the behaviour of interstitial aerosol particles. Instead, Fig. SI17 and Fig.SI18 describe better this aspect as we can analyze the time evolution of size distributions for total and interstitial aerosols modeled and observed with the Twin-inlet DMPS.

**5.20.**

**RC:** L622: "but within orders of magnitude below detection limits. . . " – I'm confused by the word "within" here, does that mean "many" orders of magnitude? Or few? I assume the latter.

**AR:** The sentence was rephrased as follows:

> ... increasing size and number concentrations during the cloud event but within one order of magnitude below detection limits...

**5.21.**

**RC:** L682: Spell out the references to field campaigns, or at least give the appropriate references.

**AR:** Names of campaigns and references have been included. Thank you.

> To this end, the models have been quite commonly compared against observations from DYCOMS II - Dynamics and Chemistry of Marine Stratocumulus II [Stevens et al., 2003] , RICO - Rain in Shallow Cumulus Over the Ocean [Rauber et al., 2007], or MPACE - Mixed-Phase Arctic Cloud Experiment [Verlinde et al., 2005] , to mention a few, that usually are based on the airborne observations over a relatively short period of time.

**5.22.**

**RC:** L683: MPACE also used longer-term ARM measurements

**AR:** The following sentence was added:

> ... that usually are based on the airborne observations over a relatively short period of time , and rarely have access to longer-term measurements like in the case of MPACE.

**5.23.**

**RC:** SL179-180: "cloud droplets and precipitation droplets" – without knowing these are different categories with different properties, it confusing why you would make this distinction.

**AR:** We have added a graphical description of the microphysical representation of size distributions of aerosol and hydrometeors in the supporting information and the manuscript was modified as shown in previous sections.

**6. TYPOGRAPHICAL COMMENTS**

**RC:** L28: At numerous locations in the manuscript the word "both" is preceded by a comma, i.e. "affecting both, the cloud optical properties and…" – remove these commas.

**AR:** Commas were removed as suggested. Thank you.

**RC:** L33-36: sentence structure does not make sense, in particular the 2nd clause ("droplet formation can be characterized…") – seems like it should be separated from the 1 st clause into its own sentence.

**AR:** You are correct. The sentence has been rephrased.

> The relative importance of aerosol concentration and updraft strength on droplet number concentration varies and depends on the local conditions droplet formation can be characterized to be aerosol or updraft limited in extreme cases, whereas typically both factors contribute. In typical atmospheric conditions, both variables drive the cloud droplet formation process, but in extreme cases distinguished as aerosol-limited regime or updraft-limited, droplet number concentrations show linear correlation just to one variable.

**RC:** L74: "…but it is…disconnected from those aerosol…"

**AR:** We reformulated the sentence as follows:

> but it is totally or partially disconnected to from those aerosol chemical effects

**RC:** L108: "Arctic" instead of "Artic"

**AR:** Thank you. The typo was removed.

**RC:** L117: "doubly periodic" instead of "doubly periodical"

**AR:** Thank you. The sentence was corrected as follows:

> Horizontal boundary conditions are  doubly periodic ...

**RC:** L125: "cloud hydrometeors"

**AR:** Thank you. The sentence was corrected as follows:

> ... aerosol particles and cloud hydrometeors...

**RC:** L141: Capitalize "In situ"

**AR:** Thank you. We added the capital letter to the title of the subsection.

**RC:** L221: "which were allowed for the second hour before the actual analysis started" – this wording is very confusing. I think you could drop the phrase "before the actual analysis started" and the meaning would be clearer.

**AR:** Thank you. The sentence was corrected as it was suggested.

> Simulations were started two hours before the beginning of the period of interest, the first hour was set as a spin-up period to allow the turbulence to develop in the absence of collision processes and drizzle formation, which were allowed for the second hour

**RC:** L276: "Figure 1 and Figure 2 [show?] the atmospheric..."

**AR:** Yes, the word "show" was forgotten. Thank you.

> Figure 1 and Figure 2 show the atmospheric boundary layer properties during both cloud events...

**RC:** L312: "Large particles ... promote drizzle formation" (not promotes)

**AR:** Thank you.

> Large particles in the sub-micron range promote drizzle formation ...

**RC:** L392: "Halo" – capitalize

**AR:** Thank you. The word Halo was corrected.

**RC:** L406: Repeat of altitude "(225 m)" – this can be removed, right?

**AR:** We meant the horizontal model domain at Puijo altitude of 225 m. We rephrased the sentence as follows:

> This procedure was carried out in every grid point through the model horizontal domain  at the Puijo altitude.

**RC:** L436: accidental "mum"

**AR:** Thank you.

**RC:** L491-492: remove )'s after Fig. 9 references

**AR:** Thank you.

> ... as reflected by Fig. 9a. Model results in Fig. 9b...

**RC:** L559: "local aerosol sources. . . raising the probability"

> local aerosol sources (i.e. heating plant, highway, residential areas) raising the probability

**RC:** L578-579: suggested rephrasing: "we decided to investigate the extent to which the modelling results. . ."

**AR:** Thank you.

> ... we decided to investigate  the extent to which the modelling results ...

**RC:** L607: I'm not sure "relevant" is the right word here to describe its importance, but I'll leave it up to you whether you want to change it

**AR:** The sentence was rephrased.

> Based on the relevant role of droplet size in the degree of modelling closure

**RC:** SL184: "non-activated"

**AR:** Thank you. The word was hyphened consistently through the manuscript and the supporting information.

**References**

J Tonttila, Z Maalick, T Raatikainen, H Kokkola, T Kühn, and S Romakkaniemi. UCLALES–SALSA v1.0: a large-eddy model with interactive sectional microphysics for aerosol, clouds and precipitation. *Geoscientific Model Development*, 10(1):169–188, 2017. . URL https://gmd.copernicus.org/articles/10/169/2017/.

R G Stevens, K Loewe, C Dearden, A Dimitrelos, A Possner, G K Eirund, T Raatikainen, A A Hill, B J Shipway, J Wilkinson, S Romakkaniemi, J Tonttila, A Laaksonen, H Korhonen, P Connolly, U Lohmann, C Hoose, A M L Ekman, K S Carslaw, and P R Field. A model intercomparison of CCN-limited tenuous clouds in the high Arctic. *Atmospheric Chemistry and Physics*, 18(15):11041–11071, 2018. . URL https://acp.copernicus.org/articles/18/11041/2018/.

J Ahola, H Korhonen, J Tonttila, S Romakkaniemi, H Kokkola, and T Raatikainen. Modelling mixed-phase clouds with the large-eddy model UCLALES–SALSA. *Atmospheric Chemistry and Physics*, 20(19): 11639–11654, 2020. . URL `https://acp.copernicus.org/articles/20/11639/2020/`.

J Tonttila, A Afzalifar, H Kokkola, T Raatikainen, H Korhonen, and S Romakkaniemi. Precipitation enhancement in stratocumulus clouds through airborne seeding: sensitivity analysis by UCLALES-SALSA. *Atmospheric Chemistry and Physics*, 21(2):1035–1048, 2021. . URL `https://acp.copernicus.org/articles/21/1035/2021/`.

Jessica Slater, Juha Tonttila, Gordon McFiggans, Hugh Coe, Sami Romakkaniemi, Yele Sun, Weiqi Xu, Pingqing Fu, and Zifa Wang. Using a coupled LES aerosol–radiation model to investigate the importance of aerosol–boundary layer feedback in a Beijing haze episode. *Faraday Discuss.*, 226(0):173–190, 2021. . URL `http://dx.doi.org/10.1039/D0FD00085J`.

J Slater, H Coe, G McFiggans, J Tonttila, and S Romakkaniemi. The effect of BC on aerosol–boundary layer feedback: potential implications for urban pollution episodes. *Atmospheric Chemistry and Physics*, 22(4): 2937–2953, 2022. . URL `https://acp.copernicus.org/articles/22/2937/2022/`.

T Raatikainen, M Prank, J Ahola, H Kokkola, J Tonttila, and S Romakkaniemi. The effect of marine ice-nucleating particles on mixed-phase clouds. *Atmospheric Chemistry and Physics*, 22(6):3763–3778, 2022. . URL `https://acp.copernicus.org/articles/22/3763/2022/`.

Silvia Calderón, Juha Tonttila, Angela Buchholz, Mika Komppula, Ari Leskinen, Hao Liqing, Dmitri Moisseev, Iida Pullinen, Petri Tiitta, Jian Xu, Annele Virtanen, Harri Kokkola, and Sami Romakkaniemi. Uclales-salsa outputs for the manuscript "aerosol-stratocumulus interactions: Towards better process understanding using closures between observations and large eddy simulations", 2022. URL `https://fmi.b2share.csc.fi/records/81a8f2f7c854465cb6b362cfdc8f19c4`.

Ming Fang, Bruce A Albrecht, Virendra P Ghate, and Pavlos Kollias. Turbulence in Continental Stratocumulus, Part II: Eddy Dissipation Rates and Large-Eddy Coherent Structures. *Boundary-Layer Meteorology*, 150(3): 361–380, 2014. ISSN 1573-1472. . URL `https://doi.org/10.1007/s10546-013-9872-4`.

Holger Siebert, Katrin Lehmann, and Manfred Wendisch. Observations of Small-Scale Turbulence and Energy Dissipation Rates in the Cloudy Boundary Layer. *Journal of the Atmospheric Sciences*, 63(5): 1451–1466, 2006. . URL `https://journals.ametsoc.org/view/journals/atsc/63/5/jas3687.1.xml`.

S Chen, L Xue, and M.-K. Yau. Impact of aerosols and turbulence on cloud droplet growth: an in-cloud seeding case study using a parcel–DNS (direct numerical simulation) approach. *Atmospheric Chemistry and Physics*, 20(17):10111–10124, 2020. . URL `https://acp.copernicus.org/articles/20/10111/2020/`.

A Bougiatioti, A Nenes, J J Lin, C A Brock, J A de Gouw, J Liao, A M Middlebrook, and A Welti. Drivers of cloud droplet number variability in the summertime in the southeastern United States. *Atmospheric Chemistry and Physics*, 20(20):12163–12176, 2020. . URL `https://acp.copernicus.org/articles/20/12163/2020/`.

Virendra P Ghate, Bruce A Albrecht, and Pavlos Kollias. Vertical velocity structure of nonprecipitating continental boundary layer stratocumulus clouds. *Journal of Geophysical Research: Atmospheres*, 115 (D13), 2010. . URL `https://agupubs.onlinelibrary.wiley.com/doi/abs/10.1029/2009JD013091`.

J Rémillard, A M Fridlind, A S Ackerman, G Tselioudis, P Kollias, D B Mechem, H E Chandler, E Luke, R Wood, M K Witte, P Y Chuang, and J K Ayers. Use of Cloud Radar Doppler Spectra to Evaluate Stratocumulus Drizzle Size Distributions in Large-Eddy Simulations with Size-Resolved Microphysics. *Journal of Applied Meteorology and Climatology*, 56(12):3263–3283, 2017a. . URL `https://journals.ametsoc.org/view/journals/apme/56/12/jamc-d-17-0100.1.xml`.

J Rémillard, A M Fridlind, A S Ackerman, G Tselioudis, P Kollias, D B Mechem, H E Chandler, E Luke, R Wood, M K Witte, P Y Chuang, and J K Ayers. Use of Cloud Radar Doppler Spectra to Evaluate Stratocumulus Drizzle Size Distributions in Large-Eddy Simulations with Size-Resolved Microphysics. *Journal of Applied Meteorology and Climatology*, 56(12):3263–3283, 2017b. . URL `https://journals.ametsoc.org/view/journals/apme/56/12/jamc-d-17-0100.1.xml`.

E Jung, B A Albrecht, H H Jonsson, Y.-C. Chen, J H Seinfeld, A Sorooshian, A R Metcalf, S Song, M Fang, and L M Russell. Precipitation effects of giant cloud condensation nuclei artificially introduced into stratocumulus clouds. *Atmospheric Chemistry and Physics*, 15(10):5645–5658, 2015. . URL `https://acp.copernicus.org/articles/15/5645/2015/`.

Hyunho Lee, Ann M Fridlind, and Andrew S Ackerman. An Evaluation of Size-Resolved Cloud Microphysics Scheme Numerics for Use with Radar Observations. Part II: Condensation and Evaporation. *Journal of the Atmospheric Sciences*, 78(5):1629–1645, 2021. . URL `https://journals.ametsoc.org/view/journals/atsc/78/5/JAS-D-20-0213.1.xml`.

I Boutle, J Price, I Kudzotsa, H Kokkola, and S Romakkaniemi. Aerosol–fog interaction and the transition to well-mixed radiation fog. *Atmospheric Chemistry and Physics*, 18(11):7827–7840, 2018. . URL `https://acp.copernicus.org/articles/18/7827/2018/`.

Mark Z. Jacobson. *Fundamentals of atmospheric modeling*. Cambridge University Press, 2005. ISBN 9780521548656. URL `http://www.cambridge.org/gb/academic/subjects/earth-and-environmental-science/atmospheric-science-and-meteorology/fundamentals-atmospheric-modeling-2nd-edition?format=PB&isbn=9780521548656`.

P Tiitta, A Leskinen, V Kaikkonen, E Molkoselkä, A Mäkynen, J Joutsensaari, S Calderon, S Romakkaniemi, and M Komppula. Intercomparison of holographic imaging and single-particle forward light scattering in-situ measurements of liquid clouds in changing atmospheric conditions. *Atmospheric Measurement Techniques Discussions*, 2022:1–20, 2022. . URL `https://amt.copernicus.org/preprints/amt-2021-423/`.

Bjorn Stevens, Donald H Lenschow, Gabor Vali, Hermann Gerber, A Bandy, B Blomquist, J L. Brenguier, C S Bretherton, F Burnet, T Campos, S Chai, I Faloona, D Friesen, S Haimov, K Laursen, D K Lilly, S M Loehrer, Szymon P Malinowski, B Morley, M D Petters, D C Rogers, L Russell, V Savic-Jovcic, J R Snider, D Straub, Marcin J Szumowski, H Takagi, D C Thornton, M Tschudi, C Twohy, M Wetzel, and M C van Zanten. Dynamics and Chemistry of Marine Stratocumulus—DYCOMS-II. *Bulletin of the American Meteorological Society*, 84(5):579–594, 2003. . URL `https://journals.ametsoc.org/view/journals/bams/84/5/bams-84-5-579.xml`.

Robert M Rauber, Bjorn Stevens, Harry T Ochs, Charles Knight, B A Albrecht, A M Blyth, C W Fairall, J B Jensen, S G Lasher-Trapp, O L Mayol-Bracero, G Vali, J R Anderson, B A Baker, A R Bandy, E Burnet, J.-L. Brenguier, W A Brewer, P R A Brown, R Chuang, W R Cotton, L Di Girolamo, B Geerts, H Gerber, S Göke, L Gomes, B G Heikes, J G Hudson, P Kollias, R R Lawson, S K Krueger, D H Lenschow,

L Nuijens, D W O'Sullivan, R A Rilling, D C Rogers, A P Siebesma, E Snodgrass, J L Stith, D C Thornton, S Tucker, C H Twohy, and P Zuidema. Rain in Shallow Cumulus Over the Ocean: The RICO Campaign. *Bulletin of the American Meteorological Society*, 88(12):1912–1928, 2007. . URL `https://journals.ametsoc.org/view/journals/bams/88/12/bams-88-12-1912.xml`.

Johannes Verlinde, Jerry Y. Harrington, G. M. McFarquhar, J. H. Mather, D. Turner, B. Zak, M. R. Poellot, T. Tooman, A. J. Prenni, G. Kok, E. Eloranta, A. Fridlind, Chad Bahrmann, K. Sassen, P. J. Demott, and A. J. Heymsfield. Overview of the mixed-phase arctic cloud experiment (m-pace). pages 4115–4120, December 2005. 85th AMS Annual Meeting, American Meteorological Society - Combined Preprints ; Conference date: 09-01-2005 Through 13-01-2005.